# Optimizing the coalition gain in Online Auctions with Greedy Structured Bandits

**Dorian Baudry**[1,2,*]    **Hugo Richard**[2,*]    **Maria Cherifa**[2,*]    **Clément Calauzènes**[2]

**Vianney Perchet**[2]

[1] Department of Statistics, University of Oxford. [2]
[2] Inria Fairplay Joint team, CREST, ENSAE Paris, Criteo AI Lab

## Abstract

Motivated by online display advertising, this work considers repeated second-price auctions, where agents sample their value from an unknown distribution with cumulative distribution function $F$. In each auction $t$, a decision-maker bound by limited observations selects $n_t$ agents from a coalition of $N$ to compete for a prize with $p$ other agents, aiming to maximize the cumulative reward of the coalition across all auctions. The problem is framed as an $N$-armed structured bandit, each number of player sent being an arm $n$, with expected reward $r(n)$ fully characterized by $F$ and $p + n$. We present two algorithms, Local-Greedy (`LG`) and Greedy-Grid (`GG`), both achieving *constant* problem-dependent regret. This relies on three key ingredients: **1.** an estimator of $r(n)$ from feedback collected from any arm $k$, **2.** concentration bounds of these estimates for $k$ within an estimation neighborhood of $n$ and **3.** the unimodality property of $r$ under standard assumptions on $F$. Additionally, `GG` exhibits problem-independent guarantees on top of best problem-dependent guarantees. However, by avoiding to rely on confidence intervals, `LG` practically outperforms `GG`, as well as standard unimodal bandit algorithms such as `OSUB` or multi-armed bandit algorithms.

## 1 Introduction

The online display advertising has seen remarkable evolution in recent decades [14, 30, 33, 25]. Publishers, who are the suppliers of digital ad space on the internet, sell display spots for ads to advertisers through real-time bidding in spot auctions, with many of these auctions being conducted using first or second-price mechanisms [20]. Due to the technological complexity of online advertising, advertisers usually delegate the task of buying ad placements to demand-side platforms (DSP) that operate many advertising campaigns. This interaction between DSP and the publisher, can be simplified as the publisher acting as multiple ad auctions selling ad impressions (online displays), while the DSP acts as a *centralized coalition*: at each time step, it determines which campaign(s) from the coalition participate to the auction to maximize their total gain. The chosen campaign(s) then compete with others to secure impressions. The primary goal of advertising companies is then to maximize the cumulative utility: the total value of impressions won minus their costs. This raises a fundamental question: *how many ad campaigns should participate in the auction to optimize the overall utility?* In the *interim* setting, where the DSP observes current bidder values before deciding, it's known that only the highest value bidder should be sent. However, online privacy enhancements in browsers necessitate *ex-ante* decisions from DSPs [8], without exact value knowledge. Here, the problem becomes challenging: choosing a small number of campaigns can make it difficult to secure impressions, while securing the spot with a large number of bidders inevitably raises the price due

38th Conference on Neural Information Processing Systems (NeurIPS 2024).

---

*Equal contribution.

[2]Corresponding author: `dorian.baudry@ox.ac.uk`

to competition. In this paper, this problem is formalized and solved via novel Multi-Armed-Bandit (MAB) algorithms.

**Problem statement**    Consider a sequence of $T$ ad impressions sold through *second price auctions* (see [20] for a survey). At auction $t \in [T]$, each participant (bidder) bids on the item based on its own (stochastic) value for the item. The highest bidder wins the item and pays a price equal to the second highest bid. The *decision maker* (the DSP) runs $N \in \mathbb{N}^*$ advertising campaigns forming a *coalition*. At time $t$, two groups of bidders participate: (1) $n_t \in [N]$ bidders from the coalition chosen by the decision maker *ex-ante* – without knowing the realization of the bidders' values – and (2) $p \in \mathbb{N}^*$ other bidders, that we call the *competition*. When a bidder from the coalition wins the auction, the decision maker observes the realized value for the winner (also called *winning bid*). In the rest of the paper, the following assumptions about the behavior of bidders is made.

**Assumption 1.** *All bidders are identical, their values are sampled i.i.d. from a distribution supported on $[0, 1]$ characterized by its cumulative distribution function (c.d.f.) $F$. All bidders bid their value.*

Assuming identical bidders with i.i.d values is a strong but widespread assumption in auction theory [20, 22], known as the symmetric bidders case. It is particularly relevant in online advertising, notably in homogeneous impression markets where advertisers compete for similar ad displays due to shared objectives, target demographics, or placement competition. The bounded support assumption is also standard, as letting an automated system bid arbitrarily large values is unrealistic. Finally, bidders bid their value as this is a weakly dominant strategy in this case. Lastly, assuming a known number of competitors $p$ is frequently seen in auction models (see for instance [20] chapter 3.2.2). Under Assumption 1, the expected reward received by the decision maker at time $t$ is given by $r(n_t)$, where $r$ is the *expected reward function*, defined by

$$r : n \in [N] \mapsto r(n) \coloneqq \mathbb{E}_{\mathbf{v}=(v_i)_{i \in [n+p]} \sim F \times \cdots \times F} \left[ (\mathbf{v}_{(1)} - \mathbf{v}_{(2)}) \mathbb{1} \left\{ \underset{i \in [n+p]}{\operatorname{argmax}} \, v_i \in [n] \right\} \right] \quad (1)$$

where $\mathbf{v}_{(1)}$ and $\mathbf{v}_{(2)}$ are respectively the first and second maximum of $\mathbf{v}$, and $[n]$ is used to abbreviate $\{1, \ldots, n\}$. The problem therefore reduces to a MAB where the decision maker chooses *arms* $n_1, \ldots, n_T \in [N]$ sequentially and aims to minimize its cumulative *expected regret* $\mathcal{R}(T)$ defined by

$$\mathcal{R}(T) = \sum_{t \leq T} r(n^*) - r(n_t), \quad \text{with} \quad n^* = \underset{n \in [N]}{\operatorname{argmax}} \, r(n), \quad (2)$$

given that privacy constraints from the browser [8] only let the decision maker observes (1) if the coalition won, (2) the realization of the maximum value when winning.

**Related works**    Following (2), the problem presented in this paper can be formulated as a Multi-Arm Bandits (MAB, see [21] for a survey). In MAB, a learner repeatedly selects from a set of actions, or "arms", each yielding a reward. The goal is to maximize total rewards by striking a balance between exploration (sampling various arms to learn their rewards) and exploitation (picking the arms with the highest anticipated rewards based on collected feedback). While the literature has known a significant development in the last years ([2, 7, 17], to name a few), the most popular approaches arguably remain *exponential weights algorithms* (`EXP3`, [4]) in adversarial settings, and *optimism in face of uncertainty* (`UCB`, [3]) when rewards are stochastic.

While `UCB` and `EXP3` can both tackle the regret minimization problem presented here, they inevitably achieve sub-optimal performance due to not using the inherent *structure* of the expected reward function. Several types of structure have been explored in the bandit literature, some notable examples being linear bandits [1], Lipschitz bandits [23], or unimodal bandits [9, 28, 29]. The problem considered here is novel in the literature of structured bandits, arising from the observability restrictions coming with privacy-enhancing systems. Still, in the next section we show that unimodality – in this paper the fact that $r$ admits only one local (hence global) maximum – is in many cases inherited from this stronger structure. A typical strategy to exploit unimodality - also used in this work - consists in playing a standard bandit policy (such as `UCB`) on a well chosen subset of arms (OSUB, [9]).

Last, the use of online learning algorithms to tackle repeated auction problems have been explored in various contexts ([26, 12, 5, 32, 27, 11]). However, none of these works approach the problem through the perspective of a coalition of bidders, and are thus not applicable to this setting.

Table 1: Comparison of regret guarantees for different algorithms

| Algorithm | Regret upper bound |
|---|---|
| EXP3 [4] | $\mathcal{O}(\sqrt{NT})$ |
| UCB1[3] | $\mathcal{O}\left(\sum_{n\in[N]} \frac{\log(T)}{\Delta_n} \wedge \sqrt{NT}\right)$ |
| OSUB [9] | $\mathcal{O}\left(\frac{\log(T)}{\Delta_{n^\star+1}} + \frac{\log(T)}{\Delta_{n^\star-1}} + \sum_{n\in[N]} \frac{\Delta_n \log\log(T)}{\Delta^2}\right)$ |
| LG (this paper) | $\tilde{\mathcal{O}}_N(\sum_{n\in[N]} \frac{\Delta_n}{\Delta^2})$ |
| GG (this paper) | $\tilde{\mathcal{O}}_N(\sum_{n\in\mathcal{B}^\star} \frac{1}{\Delta_n} + \sum_{n\in\mathcal{S}} \frac{\Delta_n}{\Delta^2}) \wedge \widetilde{\mathcal{O}}(\sqrt{(\log(N) + |\mathcal{B}^\star|)T})$ |

**Outline and contributions.** Section 3 presents two novel bandit algorithms: LG (Local Greedy) which is inspired by OSUB[9], and GG (Greedy Grid) which combines Local Greedy and a successive elimination strategy. Theorem 2 and Theorem 3 provide upper bounds on the regret of LG and GG respectively, which are summarized in Table 1. Both algorithms achieve problem-dependent regret independent of $T$. However, their scaling differs: the regret of LG depends on the *worst local gap* $\Delta = \min_{n\in[N]} |r(n+1) - r(n)|$, while for GG it only depends on the gaps $\Delta_n = r(n^*) - r(n)$. Furthermore, w.h.p. GG only suffers regret for arms in a *reference grid* $\mathcal{S}$ containing $\mathcal{O}(\log(N))$ arms and in a *neighborhood* $\mathcal{B}^\star$ of the optimal arm. All these quantities, as well as the notation $\tilde{\mathcal{O}}$ and $\widetilde{\mathcal{O}}_N$ (hiding logarithmic factors), are defined in Section 3. These regret upper bounds rely on three key ingredients presented in Section 2: (1) an estimator of $r(n)$ from feedback collected from any arm $k$ (2) novel concentration bounds on these estimates for $k$ within an estimation neighborhood of $n$ (Theorem 1) and (3) the unimodality property of $r$ under standard assumptions on $F$. Lastly, Appendix D provides an experimental benchmark comparison of the performance of GG, LG and their competitors: LG has the lowest expected regret among the algorithms tested. Indeed, LG avoids the explicit use of the confidence bounds in the algorithm which makes it more practical, even though GG admits better theoretical guarantees.

## 2 Estimating the reward function from samples of powers of $F$

In this part, we put aside the sequential nature of the repeated auction setting that we introduced and consider the problem of estimating the expected reward as a function of the number of bidders, given a stream of collected data. We first present a formulation of the expected reward function in terms of powers of the c.d.f. $F$. Then, we leverage this formula to introduce *power estimates*, as a solution to estimate the expected reward of an arm $n \in [N]$ from samples collected from an arm $k \in [N]$. Lastly, we discuss the theoretical properties of these estimates, introducing upper and lower confidence bounds on the expected reward in Theorem 1.

### 2.1 Properties of the expected reward

The expected reward function $r$ (Eq. (1)) can be expressed as a function of $n$, $p$ and the c.d.f. $F$.

**Lemma 1.** *The expected reward function defined in Equation* (1) *satisfies,*

$$n \in [N] \mapsto r(n) = n \int_0^1 F^{p+n-1}(x) - F^{p+n}(x)\mathrm{d}x \tag{3}$$

The proof can be found in Appendix A.1 and is based on properties of order statistics.

This particular definition of $r(n)$, which is a product of $n$ and a function that decreases with $n$, suggests that $r$ could be unimodal for some choices of $F$. In the rest of the paper, we restrict ourselves to distributions that guarantees unimodal reward functions.

**Assumption 2.** *$F$ and $p$ are such that the reward function $r$ in Equation* (3) *is unimodal*

As the next lemma shows, many classical distributions lead to unimodal rewards for all $p \in \mathbb{N}$.

**Lemma 2.** *Let $F$ be the cumulative distribution function of a Bernoulli, truncated exponential or Complementary Beta distribution. Then, for any $p \in \mathbb{N}^*$, $r$ in Equation* (3) *unimodal.*

The proof of Lemma 2 can be found in Appendix A.2. Note that the Complementary Beta distributions [19], chosen for technical reasons, are similar to Beta distributions and any Beta distribution can be approached by a Complementary Beta. Furthermore, in Appendix A.3 we present experiments suggesting that $r$ is unimodal for all $p \in \mathbb{N}^*$ if $F$ is the c.d.f of Beta or Kumaraswamy distributions. However, we also show in Appendix A.4 that this is not always the case, by providing a counter example. Nonetheless, we argue that (complementary) beta or truncated exponentials are flexible models for real world data, so Assumption 2 is reasonable in practice. We furthermore discuss in Section 3.2 the adaptation of our algorithms if this was not the case.

## 2.2 Estimation of powers of $F$

Consider the feedback $\overline{W}_k = (w_{k,1}, \dots, w_{k,m_k})$ gathered after playing arm $k$ and winning the auction $m_k$ times. $\overline{W}_k$, represents the sequence of first values (value of the winning bid) which has been *collected by arm $k$*.

It is well known that the marginal distribution of any order statistic can be expressed as a function of the c.d.f. $F$ (see Section 2.1 of [10]). The distribution of any element of $\overline{W}_k$ has cumulative distribution function $F_k : x \in [0,1] \to F^{k+p}(x)$, which clearly exhibits a one-to-one mapping between $F_k(x)$ and $F(x)$. Hence, given $\overline{W}_k$, for any $\ell \in \mathbb{N}$ we can estimate $F^\ell$ by

$$\widetilde{F}_{k+p}^\ell : x \mapsto (\widehat{F}_{k+p}(x))^{\frac{\ell}{k+p}}, \text{ where } \widehat{F}_{k+p} : x \mapsto \frac{1}{m_k} \sum_{j=1}^{m_k} \mathbb{1}\{w_{k,j} \leq x\} \text{ (emp. c.d.f. of } \overline{W}_k). \quad (4)$$

**Estimation of $r$**    Consider any arm $n \in [N]$. Following Equation (3), it appears that estimating both $F^{n+p}$ and $F^{n+p-1}$ is sufficient to construct an estimate of $r(n)$. According to Equation (4), this can be done from samples originated from any arm $k \in [N]$, by using the *simple estimate*

$$\widehat{r}_k(n) = n \int_0^1 \left( \widetilde{F}_{k+p}^{n+p-1}(x) - \widetilde{F}_{k+p}^{n+p}(x) \right) \mathrm{d}x . \quad (5)$$

Furthermore, it also clear that any convex combination of estimates can become a new estimate, however in the rest of the paper we focus on simple estimates for simplicity.

**Remark 1** (Adaptation to different feedback). *A similar procedure can be derived for a setting where the sequence of second prices would be observed instead. Indeed, their distribution would be $G_k : x \in [0,1] \mapsto (k+p)F(x)^{k+p-1} - (k+p-1)F(x)^{k+p}$, which can lead to a reward estimate similar to (5) by using a suitable inversion formula. The same can be said for the case where both first and second prices are observed, with additional complexity because the joint distribution should be considered since for each auction the first and second price are dependent variables.*

## 2.3 Concentration of estimates of the reward function

We now introduce the first theoretical contribution of this paper: confidence bounds on the deviations of an empirical estimate $\widehat{r}_k(n)$ w.r.t. the true expected reward $r(n)$.

**Importance of (relatively) local estimation**    In principle, (5) suggests that samples from any arm $k \in [N]$ can provide a simple estimate of the reward function of any other arm $n \in [N]$. However, we establish that the position of $k$ w.r.t. $n$ significantly impacts the concentration of $\widehat{r}_k(n)$. Intuitively, the ratio $(n+p)/(k+p)$ determines how the uncertainty on $F^{k+p}$ propagates on the reward after performing the inversion to obtain an estimate of $F^{n+p}$. Indeed, considering any $i \in \mathbb{N}$, if for some $x \in [0,1]$ the deviation $F(x)^i - \widehat{F}_i(x)$ is small then a first order approximation provides that

$$\forall j \in \mathbb{N} \; : \; (F(x)^i)^{\frac{j}{i}} - \widehat{F}_i(x)^{\frac{j}{i}} \approx (F(x)^i - \widehat{F}_i(x)) \times \frac{j}{i} F_i(x)^{\frac{j}{i}-1} . \quad (6)$$

Hence, a small error on $F(x)^i$ is multiplied by $\frac{j}{i} F_i(x)^{\frac{j}{i}-1}$ to obtain the resulting error on $F(x)^j$. For $j \geq i$ this term can be as large as $j/i$ while for $j < i$ it can be arbitrarily large if $F^i(x)$ is very small. This observation motivates a restriction on the range of arms that can be used to estimate the reward of a given arm $n$, that we call its *estimation neighborhood*. We use the convention that arms smaller than 1 or greater than $N$ exist but have not collected any sample and have a known reward of 0.

**Definition 1** (Estimation neighborhood of an arm $n$). *Assume[3] that $p \geq 4$ Then, the estimation neighborhood of $n$ is the range $\mathcal{V}(n) = [v_\ell(n), v_r(n)] = \left\{ k \in [N] : k + p \in \left[ \frac{n+p}{2}, \frac{3}{2}(n + p - 1) \right] \right\}$. We call $v_\ell(n)$ and $v_r(n)$ respectively the furthest left and right neighbor of $n$.*

**Theorem 1** (Concentration of simple estimates). *Consider any $n \in [N]$ and $k \in \mathcal{V}(n)$. Let $\widehat{r}_k(n)$ be defined according to (5) from $m_k$ samples collected by $k$. Then, there exists some constants $\beta_{k,n}$ (depending on $n, k, p$) and $\xi_{k,n,F}$ (additionally depending on $F$) such that, with probability $1 - \delta$,*

$$|\widehat{r}_k(n) - r(n)| \leq \beta_{k,n} \sqrt{\frac{\log\left(\frac{2\lceil n\sqrt{m_k}\rceil}{\delta}\right)}{m_k}} + n \times \xi_{k,n,F} \left( \frac{\log\left(\frac{2\lceil n\sqrt{m_k}\rceil}{\delta}\right)}{m_k} \right)^{\frac{n+p-1}{k+p}}. \tag{7}$$

*Furthermore, the constants admit universal upper bounds for any $n, k, p, F$. For instance if $m_k \geq 4$ it holds that $\beta_{k,n} \leq 33$ and $\gamma_{k,n,F} \leq 100$.*

*Proof sketch (see Appendix B for the detailed proof).* The first ingredient consists in approximating the reward formulation (3) by a Riemann sum: for some step size $D^{-1} > 0$, it holds that $\widehat{r}_k(n) - r(n) = \frac{n}{D} \sum_{s=0}^{D-1} \mathcal{E}(x_s) + \text{err}_D$, with $x_s = s/D$ for all $s \in \{0, \ldots, D-1\}$. In Lemma 4 we use elementary properties of $F$ to show that the approximation error satisfies $\text{err}_D \in [0, nD^{-1}]$. Next, we upper and lower bound $\mathcal{E}(x_s)$ with different concentration bounds according to the value of $F_{k,s} := F(x_s)^{k+p}$. More precisely, for any $\delta \in (0,1)$ the following bounds hold each with probability at least $1 - \delta$,

$$\begin{cases} |\widehat{F}_k(x_s) - F_{k,s}| \leq \sqrt{F_{k,s}} \times \sqrt{\frac{3\log\left(\frac{2}{\delta}\right)}{m_k}} & \text{if } F_{k,s} \in I_0 := \left[ \frac{3\log(2/\delta)}{m_k}, 1 \right] & \text{(Chernoff)}, \\ \widehat{F}_k(x_s) \leq \frac{6\log(2/\delta)}{m_k} & \text{if } F_{k,s} \in I_1 := \left( \frac{\delta}{m_k}, \frac{3\log(2/\delta)}{m_k} \right) & \text{(Chernoff)}, \\ \widehat{F}_k(x_s) = 0 & \text{if } F_{k,s} \in I_2 := \left[ 0, \frac{\delta}{m_k} \right] & \text{(union bound)}. \end{cases}$$

These results are derived in Lemma 5 from a well-known multiplicative form of the Chernoff bound for Bernoulli random variables [16]. Then, the analysis consists in using the appropriate bound for each point $s \in \{0, \ldots, D-1\}$. The interval $I_0$ provides the first term in (7), which is dominant in terms of $m_k$, and we make $\beta_{k,n}$ fully independent of $F$ by carefully using some properties of the reward function. The two remaining intervals $I_1$ and $I_2$ provide the second term in (7), and $\gamma_{k,n,F}$ depend on $F$ through the boundaries of the interval $I_1$. The corresponding factor in $\xi_{k,n,F}$ can be bounded by 1 or estimated in practice (see Appendix B.4). $\qquad \square$

In Appendix B, we give the expression of $\beta_{k,n}$ and $\xi_{k,n,F}$ and provide in (17) and (19) fully explicit upper and lower confidence bounds on $\widehat{r}_k(n)$, depending on all problem parameters, and that are much tighter than what the universal constants provided in the theorem suggest. These universal constants are purely indicative, in order to assess that $\beta_{k,n}$ and $\xi_{k,n,F}$ do not diverge for any value of the problem parameters. We now provide more high-level comments on the derivation of this result.

**Discussion** The proof of Theorem 1 is non-trivial, and the careful usage of the Chernoff bounds that we introduced is crucial to obtain tight bounds on $\widehat{r}_k(n)$ for two reasons. First, it seems necessary to concentrate estimates from arms $k > n$ (see the discussion below (6)), which are instrumental to the performance of the bandit algorithms presented in the next section. Secondly, by exhibiting powers of $F$, they make $\beta_{k,n}$ **not** increasing linearly in $n$, which is not easy to achieve. Indeed, it is clear from the analysis that this cost would be inevitable with standard Hoeffding bounds. However, completely avoiding $n$ seems difficult in general, so our proof provides a way to mitigate its cost by multiplying it by a higher power of $m_k^{-1}$, at least $m_k^{-\frac{2}{3}}$ (if $k + p = \frac{3}{2}(n + p - 1)$). This is the theoretical motivation for the definition of $\mathcal{V}(n)$ (Definition 1): while $k + p = 2(n + p - 1)$ would lead to theoretically valid results, it would not ensure that the linear term in $n$ is second-order in $m_k$.

We now conclude this section by exhibiting a condition on $F$ that allows to reduce the scaling of the confidence bound in $n$ to logarithmic terms.

---

[3] This assumption simplifies the presentation but our theoretical results can easily be adapted for $p < 4$ if $n - 1$ and $n + 1$ are included by default in $\mathcal{V}(n)$.

**Lemma 3** (Improved bound for Lipschitz quantile function). *Assume that $k \in \mathcal{V}(n)$ and $F^{-1}$ is L-Lipschitz, then there exists an absolute constant $\xi$ such that with probability $1 - \delta$ it holds that*

$$|\widehat{r}_k(n) - r(n)| \leq \beta_{k,n}\sqrt{\frac{\log\left(\frac{2\lceil n\sqrt{m_k}\rceil}{\delta}\right)}{m_k}} + \xi L \log\left(\frac{4\lceil n\sqrt{m_k}\rceil}{\delta}\right)\left(\frac{\log\left(\frac{2\lceil n\sqrt{m_k}\rceil}{\delta}\right)}{m_k}\right)^{\frac{n+p-1}{k+p}} \quad (8)$$

This result is proved in Appendix B.3, and shows that for some distributions (e.g. "close" to uniform) the confidence bounds converge relatively fast to standard sub-Gaussian type of bounds, even for very large $n$. Whether this result holds in general remains open.

# 3 Bandit algorithms

## 3.1 Bandit algorithms: Local-Greedy (`LG`) and Greedy-Grid (`GG`)

We now detail the two novel bandit algorithms proposed to tackle the problem presented in Section 1. Both rely on the use of simple estimates of $r(n)$ (see Section 2) by arms present in its *estimation neighborhood* $\mathcal{V}(n)$ (see Definition 1) and theoretically motivated by Theorem 1. In this section, for ease of exposition, we describe algorithms as if feedback was collected at every time steps. In Appendix C.1, we show that the algorithms and their guarantees only require a slight adaptation when the feedback is collected only when the auction is won.

### 3.1.1 Local-Greedy

We first present Local-Greedy (`LG`), which is a natural adaptation of a standard policy in unimodal bandits, `OSUB` [9].The main idea of `OSUB` is to play `UCB` locally around a reference arm, and eventually reach the optimal arm $n^\star$ by gradually moving the reference arm in its direction. With `LG`, we adapt this principle to efficiently exploit the structure of the problem considered: at each round $t$, `LG` defines a reference arm $\ell_t$, called *leader*, but plays *greedily* in the *neighborhood* $\mathcal{V}(\ell_t)$, based on simple power estimates computed with samples from $\ell_t$ only. In addition a *sampling requirement*, implemented by a parameter $\alpha \in (0, 1)$, is used in order to ensure the good concentration of these estimates. We detail Local-Greedy in Algorithm 1 below.

---
**Algorithm 1** Local Greedy (`LG`)

**Input:** exploration parameter $\alpha$, neighborhoods $(\mathcal{V}(n))_{n \in [N]}$ (Definition 1)
Play $n_1 = 1$ and observe $w \sim F^{1+p}$ ;                                    ▷ `Initialization`
**for** $t \geq 2$ **do**
   Set $\ell_t = n_{t-1}$, compute $(\widehat{r}_{\ell_t}(n))_{n \in \mathcal{V}(\ell_t)}$ (Eq.(5)) ;   ▷ `Compute estimates from the leader`
   **If** $m_t := |\{s \in [t-1], n_s = \ell_t\}| \leq \alpha t$: play $n_t = \ell_t$ ;    ▷ `Linear sampling requirement`
   **Else:** play $n_t \in \text{argmax}_{n \in \mathcal{V}(\ell_t)} \widehat{r}_{\ell_t}(n)$ ;              ▷ `Greedy play in` $\mathcal{V}(\ell_t)$
   Observe $w \sim F^{n_t + p}$ ;                                         ▷ `Record feedback`

---

**High-level properties of `LG`**   First, using Greedy instead of `UCB` is only possible because of the structure of the problem: when $\ell_t$ is well-explored the estimates of arms in $\mathcal{V}(\ell_t)$ computed with samples from $\ell_t$ are sufficiently close to the true reward, so that *no exploration is needed*. The sampling requirement then guarantees that all greedy plays are made when $\ell_t$ is well explored.

A second property is that since $|\mathcal{V}(\ell_t)|$ grows with $\ell_t$, a sequence of *locally optimal moves* (best play in a given neighborhood) allows to reach the optimal arm exponentially fast (in $\mathcal{O}(\log(N))$ steps), which is particularly interesting in practice if $N$ is large. On the other hand, `LG` might suffer from the inherent drawback of any "local" policy: identifying a high-rewarding arm in a neighborhood can take a long time if the reward curve in this area is flat (depending on how small are the "local" gaps). This problem can be attenuated, but not solved, by adding an initial exploration phase. We propose Greedy-Grid, presented in the next section, as a way to fully address this issue.

Lastly, requiring only the computation of empirical reward estimates is a strength of Local-Greedy. Indeed, deriving tighter confidence bounds would improve its analysis, but not the practical implementation (and performance) of the algorithm.

### 3.1.2 Greedy-grid

The concept of Greedy-Grid is very intuitive: it plays a Local-Greedy strategy only if it can tell which segment of the reward function contains the best arm with high probability. To implement this idea, `GG` uses a Successive-Elimination procedure [13] on a *subset* of arms forming a *reference grid*, denoted by $\mathcal{S}$.

**Reference grid**  The grid $\mathcal{S}$ is designed so that two of its successive arms belong to their respective neighborhood (Definition 1), and can hence mutually estimate themselves and all arms in between (Theorem 1). In particular, the optimal arm can be well-estimated at least by its two closest neighbors on the grid, so its neighborhood can be "discovered" with high probability simply by sampling the points in the grid in a round-robin fashion for a sufficiently long time.

Following Definition 1, we construct $\mathcal{S} := \{s_i\}_{i \geq 1}$ recursively: $s_1 = 1$, and for $i \geq 2$ we set $s_{i+1} = \max \{s \geq s_i : s \in [N], \ s \in \mathcal{V}(s_i), s_i \in \mathcal{V}(s)\}$. We provide an illustrative example below.

**Example 1.** *For $N = 2000$ and $p = 100$ the grid is $\mathcal{S} = \{1, 50, 123, 233, 398, 645, 1016, 1572\}$.*

Any arm $n \in [N]$ admits a left and right "neighbor in the grid", denoted respectively by $v_l^{\mathcal{S}}(n)$ and $v_r^{\mathcal{S}}(n)$ and defined by: $v_l^{\mathcal{S}}(n) = 0$ if $n < \min \mathcal{S}$, $v_r^{\mathcal{S}}(n) = N + 1$ if $n > \max \mathcal{S}$ and $(v_l^{\mathcal{S}}(n), v_r^{\mathcal{S}}(n)) = \operatorname{argmin}_{(x,y) \in \mathcal{S} \setminus \{n\} : \ n \in [x,y]} (y - x)$ otherwise . We call the "bin" of arm $n$ all arms between its left and right neighbors: $\mathcal{B}(n) = \{n \in [N], v_l^{\mathcal{S}}(n') < n < v_r^{\mathcal{S}}(n')\}$. For simplicity we use the notation $\mathcal{B}^{\star} = \mathcal{B}(n^{\star})^4$.

**Greedy-Grid**  We provide the detailed implementation in Algorithm 2 below, and now describe the general principle of the algorithm. At each round, it operates in two steps. In the first step, it decides whether to play arms on the grid $\mathcal{S}$ (play the grid, to simplify), or to focus on a specific bin (and, as we will see, *play greedy*). This choice depends on an elimination procedure: an arm $k$ in $\mathcal{S}$ should be *eliminated* for this round if their *upper confidence bound* (UCB) is smaller than the best *lower confidence bound* (LCB) among all other arms. Furthermore, if there exists an eliminated arm whose index is closer to the index $i_t^*$ of the arm with the best LCB, then the unimodality assumption implies that $k$ should also be eliminated. The set of arms not eliminated at $t$ is called $\mathcal{C}_t$ in Algorithm 2.

To compute the UCB ($U_n$) and LCB ($L_n$) of an arm $n$, we elect a leader $\ell_n$ which is the arm in $[v_l^{\mathcal{S}}(n), v_r^{\mathcal{S}}(n)]$ that was played the most in the last $t$ rounds and then compute the bounds based on $\ell_n$, using Theorem 1. We show in the proof of Theorem 3 that this procedure ensures that a linear number of samples in $t$ is used to compute the UCB and LCB of arms in $\mathcal{B}^{\star}$ with high probability.

If at least one arm is not eliminated ($\mathcal{C}_t$ is not empty), arms in $\mathcal{C}_t$ are played one after the other (Round Robin). If all arms in the grid are eliminated, `GG` plays greedily in the bin $B(i_t^*)$ of the arm with the highest LCB. The empirical reward of each arm $n \in B(i_t^*)$ is computed similarly as $U_n$ and $L_n$ using samples from the leader $\ell_n$. `GG` then plays the best empirical arm $\alpha t$ times which is the same sampling requirement as `LG`.

The careful design of Greedy-Grid prevents the main theoretical drawback of Local-Greedy: since the algorithm has a very low probability to play in a sub-optimal bin, it almost never pays "local gaps" in a sub-optimal part of the reward function. However this guarantee comes at a cost: if $n^{\star}$ is not in the grid, it will never be played until the confidence intervals shrink "sufficiently" to eliminate the entire grid. Hence, `GG` might be more conservative than `LG` in practice, while offering better theoretical guarantees. We express this trade-off in the next section.

### 3.2  Regret upper bounds

We now present the theoretical results obtained for the two algorithms presented in Section 3. We first establish the regret bounds and sketch their proofs, before discussing and comparing the results. We introduce some notation, that considerably simplifies the presentation of the results.

**Notation: $\widetilde{\mathcal{O}}$ and $\widetilde{\mathcal{O}}_n$**  For any $x > 0$, we use the notation $\widetilde{\mathcal{O}}(x)$ to describe a quantity that scales in $x$, up to logarithmic terms **in $x$ and** $N$ (hence the notation is linked to the problem). Furthermore,

---

[4] We assume a unique optimal arm for simplicity, but the analysis holds if several successive arms are optimal.

---

**Algorithm 2** Greedy Grid

---

**Input:** Grid $\mathcal{S}$, confidence levels $(\delta_t)_{t \in \mathbb{N}}$, sampling parameter $\alpha$
Play $n_1 = \min \mathcal{S}$ and observe $w \sim F^{n_1 + p}$
**for** $t \geq 2$ **do**

    $\forall n \in [N] \; \ell_n = \operatorname{argmax}_{k \in [v_l^S(n), v_r^S(n)]} \overbrace{\left| \{u \in [t-1], n_u = k\} \right|}^{m_k}$ ;     ▷ Elect leaders

    $\forall n \in [N], \; L_n = \widehat{L}_{\ell_n}(n, \delta_t)$ and $U_n = \widehat{U}_{\ell_n}(n, \delta_t)$ ;     ▷ Compute UCB (21), LCB (22)

    $i_t^* = \operatorname{argmax}_{n \in [N]} L_n$ ;     ▷ Compute best lower bound index

    $\mathcal{C}_t = \{a \in \mathcal{S} : \forall s \in [N] \text{ s. t. } a \leq s \leq i_t^* \text{ or } a \geq s \geq i_t^*, U_s \geq L_{i_t^*}\}$ ;   ▷ Non-elim. grid arms

    **if** $n_{t-1} \in B(i_t^*)$ *and* $m_{n_{t-1}} \leq \alpha t$ **then**     ▷ Ensure linear sampling for bin plays
        Play $n_t = n_{t-1}$

    **else**     ▷ Play grid if non-empty or greedy in the best LCB's bin
        **If** $\mathcal{C}_t = \varnothing$: Play Round Robin on $\mathcal{C}_t$; **Else** play $\operatorname{argmax}_{n \in B(i_t^*)} \hat{r}_{\ell_n}(n)$

    Observe $w \sim F^{n_t + p}$ ;     ▷ Record feedback

---

for $n \in [N]$ we also use $\widetilde{\mathcal{O}}_n$ as a shorthand notation for $\widetilde{\mathcal{O}}(\{n^6 \vee x\} \wedge n^2 x)$. This type of constants emerges from using (7) (Theorem 1) in the analysis. Indeed, we proved that the simple estimate of an arm $n$ by an arm $k \in \mathcal{V}(n)$ admit sub-Gaussian ("square-root") confidence intervals, independent of $n$, when the sample size of $k$ is larger than $\Omega(n^6)$.

**Theorem 2** (Regret bound for Local-Greedy). *Let* $\Delta := \min_{n \in [N-1]} |r(n+1) - r(n)|$ *(worst local gap). Under Assumption 2 and with* $\alpha = (\log_{3/2} N + 1)^{-1}$*, the regret of* LG *is upper bounded by a* **problem-dependent constant**: *there exists* $(C_n)_{n \in [N] \setminus \{n^\star\}}$*, each satisfying* $C_n = \widetilde{\mathcal{O}}_N \left( \frac{\Delta_n}{\Delta^2} \right)$*, such that* $\mathcal{R}_T \leq \sum_{n \in [N] \setminus n^\star} C_n$.

*Additionally, if the arm set forms a single estimation neighborhood, that is* $\forall n \in [N] : \mathcal{V}(n) \supset [N]$*, then each constant* $C_n$ *can be refined to* $\widetilde{\mathcal{O}}_n \left( \Delta_n^{-1} \right)$*, providing* $\mathcal{R}_T = \widetilde{\mathcal{O}}(\sqrt{NT})$*, which holds even when the reward function is not unimodal.*

*Proof sketch (see Appendix C.3 for the detailed proof).* We start by the case where the arm set forms a single neighborhood. Since LG is guaranteed that any arm it selects will provide an estimate for all the other arms, this context is very similar to a full information scenario. This explains why GG achieves both constant regret depending on the gaps, and a gap-independent bound in $\sqrt{NT}$. Furthermore, the hidden logarithmic constants come from carefully using Theorem 1 to separate the linear term in $n$ from the gaps when they are small.

The general case presents an additional complexity. Indeed, it is possible that playing arm $n \neq n^\star$ is *locally optimal*, if $n$ is the best arm in the neighborhood of the current leader: playing $n$ in that context would not be unlikely. To tackle that scenario, we prove that pulling arm $n$ at time $t$ necessarily implies a *locally sub-optimal play*, in some estimation neighborhood, at some point in the past (maximized by the chosen value of $\alpha$). We then show that this cannot happen after some deterministic time w.h.p., leading to constant regret. However, since the sub-optimal play might be any arm the constant now depends on the *worst local gap* $\Delta^2$. □

**Theorem 3** (Regret upper bound for Greedy-Grid). *Suppose that* GG *is tuned with confidence level* $\delta_t = \frac{1}{N^2 t^3}$*, and* $\alpha = 1/4$*. Then, for any* $T \in \mathbb{N}$ *it holds that*

$$\mathcal{R}_T = \widetilde{\mathcal{O}}_N \left( \sum_{n \in \mathcal{B}^\star} \frac{1}{\Delta_n} + \sum_{n \in \mathcal{S}} \frac{\log(T)}{\Delta_n} \wedge \Delta_n \left( \frac{\mathbb{1}\{n < n^\star\}}{\Delta_{v_l(n^\star)}^2} + \frac{\mathbb{1}\{n > n^\star\}}{\Delta_{v_r(n^\star)}^2} \right) \right) .$$

*Additionally, it holds that* $\mathcal{R}_T = \widetilde{\mathcal{O}} \left( \sqrt{(K + |\mathcal{B}^\star|)T} \right)$*, for* $K = \lfloor \log_{3/2}(N) \rfloor$.

*Proof sketch (see Appendix C.4 for the detailed proof).* First we prove that, w.h.p., during a linear time range in $t$ GG either played the grid or in $\mathcal{B}^\star$. Hence, arms $n \in [N] \setminus \{\mathcal{S} \cup \mathcal{B}^\star\}$ are played a (universal!) constant number of times by GG in expectation. Then, for $n \in \mathcal{S}$ the term in $\frac{\log(T)}{\Delta_n}$ comes from the standard analysis of UCB [3]; while the constant bound comes from exploiting that after

a constant time $n$ the LCB of $n^\star$ eliminates its neighboors w.h.p., and by extension the entire grid. Finally, the constant bound $n \in \mathcal{B}^\star$ is derived similarly as the first bound of Theorem 2. $\qquad\square$

**Discussion**  First, we show that being able to estimate $r(n)$ from the feedback obtained after playing an arm $k$ in its estimation neighborhood leads to a regret independent of $T$ for both LG and GG. For the former, the bound depends in general on the worst *local gap* $\Delta$, while for the latter only the actual gaps $\Delta_n$ (with $n^\star$) are involved. This difference permits to obtain a problem-independent guarantee for GG for any configuration of $p$ and $N$. Furthermore, its scaling $\sqrt{K + |\mathcal{B}^*|} \le \sqrt{2n^* + \lfloor \log_{3/2}(N) \rfloor}$ can be much smaller than $\sqrt{N}$ if $n^*$ is small.

Then, we would like to discuss the impact of the concentration bound presented in Theorem 1 on the regret of both GG and LG. Indeed, a naive approach with Hoeffding bounds would not allow to remove $n$ from the first order term of the concentration bound, because of the multiplicative factor $n$ in the definition of $r(n)$. A feature of our concentration bound is that the linear scaling in $n$ does not appear in the first order term. Informally, this allows to exhibit terms of order $\widetilde{\mathcal{O}}_N(\Delta_n^{-1})$ in the regret analysis instead of $\widetilde{\mathcal{O}}(N^2\Delta_n^{-1})$, which can be significantly better for small gaps. A remark here is that the size of the grid in GG could be optimized as a larger grid makes the second order term in Theorem 1 smaller but is paid linearly in the regret.

We nevertheless highlight some potential for improvement in the analysis of LG. First, the local gaps $\Delta$ in the bound of LG could be replaced by (in spirit, referring to $\mathcal{S}$ for simplicity) $\min_{n \in [N]} |r(v_l^S(n)) - r(v_r^S(n))|$. It is clear though that this gap remains "local" and can be arbitrarily smaller than $\Delta_n$ for some arms $n \in [N]$, so the general interpretation of the results would be unchanged. Second, for simplicity, the analysis of LG was carried out using the constant upper bound of $\beta_{k,n}$ and $\xi_{k,n,F}$ but a tighter analysis could lead to a better dependency with respect to $N$.

We now justify the use of simple estimates in GG and LG. In practice, combining estimates would allow to use more samples for the estimation. However, this would make the algorithm slower, and we believe that the sampling requirement implemented in the algorithms makes the use of simple estimates efficient: potential uniform exploration in a neighborhood is replaced by a focus on a single arm, but the same quality of information is accrued. Furthermore, from a theoretical perspective union bounds over the samples collected by each arm might also cost a factor $N$ in the analysis.

Lastly, while GG admits better theoretical guarantees, LG might be more appealing in practice because it does not require to explicitly compute confidence intervals. This means that the regret bounds provided for LG are conservative, and might be refined with tighter confidence bounds without changing the algorithm.

**Adaptation for non-unimodal rewards**  While LG relies heavily on Assumption 2, GG can be readily adapted to handle non-unimodal reward functions. This is done by modifying the definition of the set of non-eliminated grid arms $\mathcal{C}_t$ to $\{s \in \mathcal{S}, U_s \ge L_{i_t^*}\}$ in Algorithm 2. In that case, the algorithm can no longer eliminate arms on the grid based on the elimination of other arms. This naturally induces that the number of plays of sub-optimal arms is no longer bounded by a constant. In Theorem 4 (see Appendix), we show that only the $\mathcal{O}(\log(T))$ term persists for $n \in \mathcal{S}$ in Theorem 3, while the problem-independent bound remains unchanged. Although we believe unimodality is necessary for achieving constant regret, this result demonstrates that, even without that assumption, GG can still provide the same logarithmic regret guarantees as UCB. However, it does so on a $|\mathcal{S}|$-armed bandit, rather than an $N$-armed bandits with $|\mathcal{S}| = \mathcal{O}(\log(N)) \ll N$ for large $N$.

**Experimental results**  In Appendix D we present a benchmark of LG, GG, UCB, EXP3 and OSUB on synthetic data in terms of the expected regret $\mathcal{R}(T)$. This benchmark illustrates the strong performance of LG relative to the other approaches. Although GG offers more robust theoretical guarantees, particularly with sub-linear problem-independent bounds, LG proves to be more effective in practice. Several factors may explain this gap between theoretical guarantees and empirical performance. First, as discussed in the previous section, the worst-case local gap in the analysis of Local Greedy (Theorem 2) might be overly conservative. This worst-case scenario could occur under a combination of unfavorable conditions, such as poor initialization far from the optimal arm and a flat reward function, paired with bad luck in exploration. However, such a scenario is likely rare in practice and was not encountered in our experiments. Additionally, Local Greedy benefits from scenarios

where it starts playing in the optimal neighborhood only after a few steps, a situation GG cannot exploit due to its need for sufficient statistical evidence to eliminate all suboptimal neighborhoods. While GG's caution leads to stronger theoretical guarantees, this comes at the cost of empirical performance. Moreover, GG's results are tied to the tightness of the confidence intervals in Theorem 2, a limitation that does not apply to LG. An interesting and challenging open problem remains whether LG can be modified to achieve the same theoretical guarantees as GG without sacrificing its performance. We leave this question for future work.

## 4    Conclusion

The bandit problem studied in this work is structured since playing arm $n$ gives a reward $r(n)$ determined by $n, p$ and the unknown c.d.f $F$ and with probability $\frac{n}{n+p}$ an observation of a sample of the distribution with c.d.f $F^{n+p}$.

While traditional bandit approaches give problem dependent bounds depending on $T$, algorithms GG and LG presented in this work have constant problem dependent bounds. Furthermore, GG and LG avoid a quadratic dependency in $N$ for large $T$ thanks to new concentration bounds introduced in Theorem 1. Overall, while GG has the best theoretical guarantees, LG has better constants and is therefore better suited for most practical problems (see the discussion at the end of Section 3 and experimental results in Appendix D).

Whether an algorithm that has the theoretical guarantees of GG and the practical performance of LG can be designed is an interesting question. We believe that the main leverage to improve the practical performance of GG might be to derive tighter concentration bounds. Possible directions to improve over Theorem 1 might include: further refining the decomposition of the integral in (3) according to the value of $F$, further use "empirical" components (depending on estimates of $F$), or even using ideas from the proof of the DKW inequality [24] to avoid the union bounds over the points of each interval in the decomposition. We leave these directions for future work.

To conclude, since in practice, a DSP can launch campaigns through multiple auctions, an interesting question is whether the current analysis could be extended to the case of $A$ auctions where a play at time $t$ is $(n_{a,t})_{a \in [A]}$ where $\sum_{a \in [A]} n_{a,t} = N$ and the reward is $\sum_{a \in [A]} r_a(n_{a,t})$ with $r_a$ determined by integers $p_a$, $n_{a,t}$ and $F_a$ in the same way that $r$ depends on $p, n_t$ and $F$. How to explore each auction in parallel in an efficient manner and how to handle the case where some auctions must be assigned zero players are then the main questions to solve.

## Acknowledgments

Dorian Baudry thanks the support of the French National Research Agency: ANR-19-CHIA-02 SCAI, ANR-22-SRSE-0009 Ocean, and ANR-23-CE23-0002 Doom. Dorian Baudry was partially funded by UK Research and Innovation (UKRI) under the UK government's Horizon Europe funding guarantee [grant number EP/Y028333/1].

Vianney Perchet's research was supported in part by the French National Research Agency (ANR) in the framework of the PEPR IA FOUNDRY project (ANR-23-PEIA-0003) and through the grant DOOM ANR-23-CE23-0002. It was also funded by the European Union (ERC, Ocean, 101071601). Views and opinions expressed are however those of the author(s) only and do not necessarily reflect those of the European Union or the European Research Council Executive Agency. Neither the European Union nor the granting authority can be held responsible for them.

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

# A Properties of the expected reward function

In this appendix we prove the results presented in Section 2.1 of the paper, and discuss the shape of the expected reward.

## A.1 Proof of Lemma 1

**Lemma 1.** *The expected reward function defined in Equation* (1) *satisfies,*

$$n \in [N] \mapsto r(n) = n \int_0^1 F^{p+n-1}(x) - F^{p+n}(x)\mathrm{d}x \tag{3}$$

*Proof.* Given $\mathbf{v} = (v_i)_{i \in [n+p]} \sim F \times \cdots \times F$, we have

$$
\begin{aligned}
r(n) &= \mathbb{E}\left[(\mathbf{v}_{(1)} - \mathbf{v}_{(2)})\mathbb{1}\left\{\operatorname*{argmax}_{i \in [n+p]} v_i \in [n]\right\}\right] \\
&\stackrel{(i)}{=} \mathbb{E}\left[(\mathbf{v}_{(1)} - \mathbf{v}_{(2)})\right]\mathbb{E}\left[\mathbb{1}\left\{\operatorname*{argmax}_{i \in [n+p]} v_i \in [n]\right\}\right] \\
&\stackrel{(ii)}{=} \left(\mathbb{E}\left[\mathbf{v}_{(1)}\right] - \mathbb{E}\left[\mathbf{v}_{(2)}\right]\right) \times \frac{n}{n+p} \\
&= \frac{n}{n+p} \times \int_0^1 \mathbb{P}(\mathbf{v}_{(1)} > x) - \mathbb{P}(\mathbf{v}_{(2)} > x)\mathrm{d}x \\
&= \frac{n}{n+p} \times \int_0^1 \mathbb{P}(\mathbf{v}_{(2)} \le x) - \mathbb{P}(\mathbf{v}_{(1)} \le x)\mathrm{d}x \\
&\stackrel{(iii)}{=} \frac{n}{n+p} \times \int_0^1 ((n+p)F^{n+p-1}(x) - (n+p-1)F^{n+p}(x) - F^{n+p}(x))\mathrm{d}x \\
&= n \int_0^1 (F^{n+p-1}(x) - F^{n+p}(x))\mathrm{d}x \;.
\end{aligned}
$$

The first equality is the definition of $r(n)$ in Equation (1). Equality $(i)$ follows by independence of the index of the maximum and the value of the maximum and second maximum. This is itself a consequence of the fact that the values are i.i.d.. Then equality $(ii)$ follows since the distribution of the index of the maximum is uniform over $n + p$. This is also a consequence of the fact that the values are i.i.d. Lastly, equality $(iii)$ follows from [10] (Equation 2.1.3) where for $k \in \{1, 2\}$, it is shown that

$$\mathbb{P}(\mathbf{v}_{(k)} \le x) = \sum_{i=n+p-k+1}^{n+p} \binom{n+p}{i}(1 - F(x))^{n+p-i}F(x)^i,$$

and the proof is concluded by substitution. $\square$

## A.2 Proof of Lemma 2

As a preliminary, we formally define the non-usual distributions considered in Lemma 2.

**Truncated exponential distribution** Let $a > 0$ be some parameter. Then, we define a truncated exponential distribution of parameter $a$ as the distribution with c.d.f. $F : x \mapsto \frac{1-e^{-ax}}{1-e^{-a}}$. Hence, $F(0) = 0$ and $F(1) = 1$, and the density of this distribution is the same as the density of the exponential distribution with same parameter on the segment $[0, 1]$, up to a normalization constant.

**Complementary Beta distribution**

**Lemma 2.** *Let $F$ be the cumulative distribution function of a Bernoulli, truncated exponential or Complementary Beta distribution. Then, for any $p \in \mathbb{N}^*$, $r$ in Equation* (3) *unimodal.*

*Proof.* We consider each family of distributions separately.

**Bernoulli distributions**  If $F$ is the c.d.f. of $\mathcal{B}(q)$ (a Bernoulli distribution of parameter $q$), then $r(n)$ is equal to the probability that exactly one player from the coalition draws a value of 1, and every other player draw a value of 0. Hence, we obtain that $r(n) = nq(1-q)^{n+p-1}$, which is trivially unimodal and maximized in $n^\star = \frac{-1}{\log(1-q)} \vee 1$, regardless of the size of the competition.

**Truncated exponential distributions**  Let $a > 0$ be the parameter of the distribution. Let $Q(x)$ be the inverse function of $F$ (the quantile function), defined by $Q(x) = \frac{1}{a} \log\left( \frac{1}{1-x(1-e^{-a})} \right) = \frac{1}{a} \sum_{k=1}^{+\infty} \frac{x^k (1-e^{-a})^k}{k}$.

Let's denote by $q(x)$ the derivative of $Q(x)$, denoted by $q(x) = \sum_{k=0}^{+\infty} \lambda_k x^k$ where $\lambda_k = \frac{1}{a}(1-e^{-a})^k$. Introducing theses functions allows us to rewrite $r(n)$ as follows,

$$
\begin{aligned}
r(n) &= n \int_0^1 F(v)^{p+n-1}(1-F(v))\mathrm{d}v \\
&= n \int_0^1 x^{p+n-1}(1-x)q(x)\mathrm{d}x \qquad \text{using } F(v) = x \\
&= n \int_0^1 x^{p+n-1}(1-x)\left( \sum_{k=0}^{+\infty} \lambda_k x^k \right) \mathrm{d}x \\
&= \sum_{k=0}^{+\infty} \lambda_k \left( \frac{n}{p+n+k} - \frac{n}{p+n+k+1} \right) \\
&= \frac{n}{n+p}\lambda_0 + n \sum_{j=1}^{+\infty} \frac{1}{n+p+j}(\lambda_j - \lambda_{j-1}) \\
&= \lambda_0(1 - \frac{p}{n+p}) + \sum_{j=1}^{+\infty}(1 - \frac{p+j}{n+p+j})(\lambda_j - \lambda_{j-1}) \\
&= \lambda_0\left( -\frac{p}{n+p} + \sum_{j=1}^{+\infty} \underbrace{\frac{\lambda_{j-1} - \lambda_j}{\lambda_0}}_{\theta_j} \frac{p+j}{n+p+j} \right)
\end{aligned}
$$

where the last inequality follows since $\lim_{j\to\infty} \lambda_j = 0$. Remark that $\theta_j \geq 0$ since $\lambda_j$ is decreasing and $\sum_{j=1}^{\infty} \theta_j = 1$.

The derivative of $r(n)$ is given by,

$$
\begin{aligned}
r'(n) &= \lambda_0\left( \frac{p}{(n+p)^2} - \sum_{j=1}^{+\infty} \theta_j \frac{p+j}{(n+p+j)^2} \right) \\
&= \lambda_0\left( \Theta_p(n) - \sum_{j=1}^{\infty} \theta_j \Theta_{p+j}(n) \right) \\
&= \lambda_0 \Theta_p(n)\left( 1 - \sum_{j=1}^{\infty} \theta_j \Gamma_{p,p+j}(n) \right) \qquad \text{where } \Gamma_{p,p+j}(n) = \frac{\Theta_{p+j}(n)}{\Theta_p(n)}
\end{aligned}
$$

the functions $\Gamma_{p,p+j}(n)$ are non-decreasing hence it is the same for their convex combination. As $\mathrm{sign}(r'(n)) = \mathrm{sign}(1 - \sum_{j=1}^{\infty} \theta_j \Gamma_{p,p+j}(n))$ it follows that $\mathrm{sign}(r'(n))$ is decreasing meaning $r(n)$ is unimodal.

**Complementary beta distributions**  Using the same change of variable as in the previous proof $(Y = F(X))$, we express the reward as follows,

$$
r(n) = n \times \mathbb{E}_{Y \sim Q}\left[ Y^{n+p-1}(1-Y) \right] ,
$$

where $Q$ denotes the quantile function associated with c.d.f $F$ (i.e. $F^{-1}$). By definition of $F$, $Y$ follows a Beta distribution of parameters $(a, b)$. We can thus compute the expected reward by using

the explicit formula for moments of the Beta distribution,

$$\mathbb{E}_{X \sim B(a,b)}[X^{n+p-1} - X^{n+p}] = \prod_{k=0}^{n+p-2} \frac{a+k}{a+b+k} - \prod_{k=0}^{n+p-1} \frac{a+k}{a+b+k}$$

$$= \prod_{k=0}^{n+p-2} \frac{a+k}{a+b+k} \times \left(1 - \frac{a+n+p-1}{a+b+n+p-1}\right)$$

$$= \prod_{k=0}^{n+p-2} \frac{a+k}{a+b+k} \times \frac{b}{a+b+n+p-1} .$$

Thanks to this expression, we prove the unimodality by analyzing the ratio $\frac{r(n+1)}{r(n)}$, that we first write as

$$\frac{r(n+1)}{r(n)} = \frac{n+1}{n} \times \frac{a+n+p-1}{a+b+n+p-1} \times \frac{a+b+n+p-1}{a+b+n+p}$$

$$= \frac{n+1}{n} \times \frac{a+n+p-1}{a+b+n+p} ,$$

and then obtain that this ratio is larger than 1 if and only if

$$(n+1)(a+n+p-1) \geq n(a+b+n+p) \iff n(a+n+p) + a + p - 1 \geq n(a+n+p) + bn$$

$$\iff n \geq \frac{a+p-1}{b} ,$$

which concludes the proof by showing the unimodality and expressing the value of the critical point.

$\square$

The proof of Lemma 2 highlights that the unimodality assumption is satisfied as soon as the quantile function, expressed as a power series, has its coefficients that slowly decrease (indeed, the $k$-th coefficient just needs to be smaller than $1 - \frac{1}{k}$ times the $k-1$-th one).

Similarly, the second proof technique highlights (up to standard algebraic manipulations) that unimodularity is guaranteed as soon as the function $n \mapsto 1 - E[X^{n+p-1}]/E[X^{n+p-2}]$ is log-concave.

### A.3 Additional discussion on the unimodality of $r$

In this section, we plot the shape of $r(n)$ for some additional families of distribution that we conjecture to be unimodal from the plots.

**Beta distribution**   The following figure, illustrate the unimodal shape of $r(n)$ for different parameters for the Beta distribution and $p$.

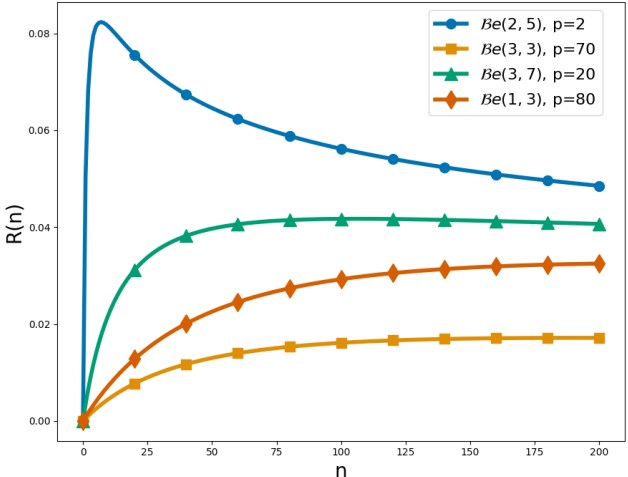

Figure 1: Shape of $r(n)$ when $F$ is Beta

**Kumaraswamy distribution** The cumulative distribution is defined by $F(x) = 1 - (1 - x^a)^b$ for some parameters $(a, b)$ (we use the notation $K(a, b)$). The following figure, illustrate the unimodal shape of $r(n)$ for different parameters of $K(a, b)$ and $p$.

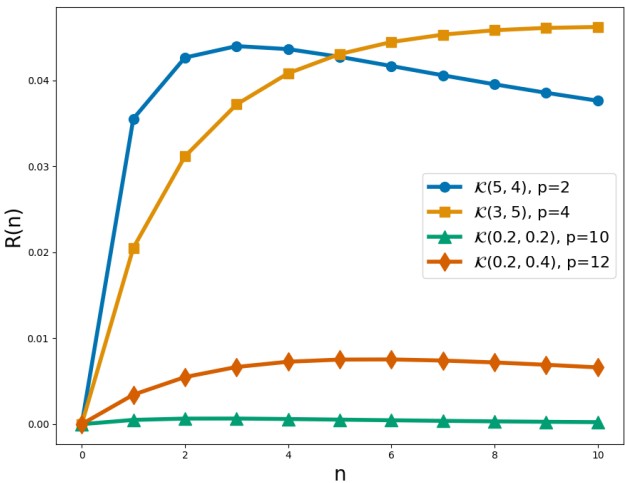

Figure 2: Shape of $r(n)$ when $F$ is Kumaraswamy distribution

We now provide and discuss an example where Assumption 2 is not satisfied.

### A.4   An example of distribution with non-unimodal rewards

Let us consider a discrete distribution supported on $\{0, 0.5, 1\}$. The counter-example emerges from putting all the probability mass in $0.5$: let us consider a small $\epsilon > 0$, identify $F$ with $\{\epsilon, 1 - \epsilon, 1\}$ and assume that there is no competition ($p = 0$). Then, we can verify with (3) that

$$r(n) = \frac{n}{2}(\epsilon^{n-1} + (1 - \epsilon)^{n-1} - (\epsilon^n + (1 - \epsilon)^n))$$
$$= \frac{n}{2}(\epsilon^{n-1}(1 - \epsilon) + \epsilon(1 - \epsilon)^{n-1})$$

Consider $\epsilon = 0.15$, we get up to a precision $0.001$ the values:

$$(r(n))_{n=1}^{7} = (0.5, 0.255, 0.191, 0.190, 0.197, 0.200, 0.198)$$

showing that $r(n)$ is not unimodal in this case.

# B Concentration bounds on simple reward estimates

## B.1 Auxiliary results

Before presenting the proof of the theorem, we present two auxiliary results that are essential to its development.

### B.1.1 Riemann sum approximation of the expected reward

The first result consists in upper bounding the deviation of a Riemann sum approximation of $r(n)$ (for some $n \in [N]$) with respect to its exact integral formulation. This result is also of practical interest, since it can prevent computing exact integrals at each step of the algorithms without altering their theoretical guarantees with an appropriate tuning of the approximation error.

**Lemma 4** (Riemann sum approximation of $r(n)$). *Let $n \in [N]$, $D \in \mathbb{N}$, and define the grid* $(x_s)_{s \in \{0, \dots, D-1\}} = \{0, \frac{1}{D}, \dots, \frac{D-1}{D}\}$. *Then, the expected reward approximation*

$$\widetilde{r}(n) = n \times \frac{1}{D} \sum_{s=0}^{D-1} \left\{ F(x_s)^{n+p-1} - F(x_s)^{n+p} \right\}$$

*satisfies*

$$|r(n) - \widetilde{r}(n)| \leq \frac{n}{D} \ .$$

*Proof.* For any $j \in \mathbb{N}$ we consider

$$S_j = \frac{1}{D} \sum_{s=0}^{D-1} F^j(x_s) \quad \text{as an approximation of} \quad I_j = \int_0^1 F^j(x) \mathrm{d}x \ .$$

We recall that since $F^j$ is a c.d.f., it is monotone, increasing and satisfies $F^j(0) = 0$ and $F^j(1) = 1$. This means that for any $s \in \{0, \dots, D-1\}$, it holds that $\forall x \in [x_s, x_{s+1}], F^j(x_s) \leq F^j(x) \leq F^j(x_{s+1})$. The linearity of the integral first provides that

$$I_j = \int_0^1 F^j(x) \mathrm{d}x = \sum_{s=0}^{D-1} \int_{\frac{s}{D}}^{\frac{s+1}{D}} F^j(x) \mathrm{d}x$$

Then, using this decomposition and the monotony of $F^j$ we obtain that

$$
\begin{aligned}
S_j \leq \frac{1}{D} \sum_{s=0}^{D-1} F^j\left(\frac{s}{D}\right) \leq \int_0^1 F^j(x)\mathrm{d}x &\leq \frac{1}{D} \sum_{s=0}^{D-1} F^j\left(\frac{s+1}{D}\right) \\
&\leq \frac{1}{D} \sum_{s=0}^{D-1} F^j\left(\frac{s}{D}\right) + \frac{1}{D} \sum_{k=0}^{D-1} \left( F^j\left(\frac{s+1}{D}\right) - F^j\left(\frac{s}{D}\right) \right) \\
&\leq \frac{1}{D} \sum_{s=0}^{D-1} F^j\left(\frac{s}{D}\right) + \frac{F^j(1) - F^j(0)}{D} \\
&= S_j + \frac{1}{D} \ .
\end{aligned}
$$

Therefore, we obtained that for any $j \in \mathbb{N}$ it holds that $S_j \leq I_j \leq S_j + \frac{1}{D}$. We conclude by using this result after splitting the reward as a difference of two integrals that can be expressed in this form, respectively with $j = n + p - 1$ and $j = n + p$. □

### B.1.2 Chernoff bounds for Bernoulli random variables

In the following lemma, we summarize the different concentration bounds that we use in the proof of Theorem 1.

**Lemma 5** (Mutltiplicative Chernoff bounds). *Let $\widehat{\mu}_m$ be the empirical average of $m$ i.i.d. Bernoulli random variables $X_1, \ldots, X_m$ with expectation $\mu$. Then, for any $\delta > 0$, each of the following bounds holds with probability at least $1 - \delta$,*

$$
\begin{cases}
|\widehat{\mu}_m - \mu| \leq \sqrt{\mu} \times \sqrt{\frac{3\log\left(\frac{2}{\delta}\right)}{m}} & \text{if } \mu \in I_0 := \left[\frac{3\log(2/\delta)}{m}, 1\right], \\
\mu_m \leq \frac{6\log(2/\delta)}{m} & \text{if } \mu \in I_1 := \left(\frac{\delta}{m}, \frac{3\log(2/\delta)}{m}\right), \\
\mu_m = 0 & \text{if } \mu \in I_2 := \left[0, \frac{\delta}{m}\right].
\end{cases}
$$

*Proof.* We first tackle the case $\mu \in I_2$, where the bound is obtained by remarking that $\mathbb{P}(\mu_m > 0) \leq \mathbb{P}(\exists i \in [m] : X_i = 1) \leq m\mu \leq \delta$. The two other cases are obtained by using the multiplicative form of the well-known Chernoff bounds [16], that provide that

$$
\forall \gamma > 0, \quad \mathbb{P}(\widehat{\mu}_m \geq (1 + \gamma)\mu) \leq e^{-m\frac{\gamma^2 \mu}{2+\gamma}} \quad, \text{ and}
$$

$$
\forall \gamma \in [0, 1], \ \mathbb{P}(\widehat{\mu}_m \leq (1 - \gamma)\mu) \leq e^{-m\frac{\gamma^2 \mu}{2}} .
$$

When considering $\gamma \leq 1$ we can further write that

$$
\mathbb{P}(|\widehat{\mu}_m - \mu| \geq \gamma\mu) \leq 2e^{-m\frac{\gamma^2 \mu}{3}} .
$$

On the other hand, for $\gamma \geq 1$ the bound for the lower deviation is trivially $0$ while the bound for the upper deviation can be written as follows,

$$
\gamma \geq 1 \Rightarrow \mathbb{P}(\widehat{\mu}_m \geq (1 + \gamma)\mu) \leq e^{-m\frac{\gamma^2 \mu}{2+\gamma}} \leq e^{-m\frac{\gamma^2 \mu}{3\gamma}} = e^{-m\mu\frac{\gamma}{3}} .
$$

Hence, the case separation between the intervals $I_0$ and $I_1$ simply consist in identifying the value $\mu$ for which a probability $1 - \delta$ can be obtained by setting an appropriate $\gamma \in [0, 1]$ or for $\gamma > 1$ in the above inequalities. More precisely, inverting the first bound provides $\gamma = \mu^{\frac{-1}{2}}\sqrt{\frac{3\log\left(\frac{2}{\delta}\right)}{m}}$, which is valid only if $\mu^{\frac{-1}{2}}\sqrt{\frac{3\log\left(\frac{2}{\delta}\right)}{m}} \leq 1 \Rightarrow \mu \geq \frac{3\log\left(\frac{2}{\delta}\right)}{m}$. This leads to the first confidence interval when $\mu \in I_0$. The same procedure for $\mu \in I_1$ leads to $\gamma = \mu^{-1}\frac{3\log\left(\frac{2}{\delta}\right)}{m}$, which provides the result stated in the lemma. This concludes the proof. $\qquad\square$

### B.2 Proof of Theorem 1

**Theorem 1** (Concentration of simple estimates). *Consider any $n \in [N]$ and $k \in \mathcal{V}(n)$. Let $\widehat{r}_k(n)$ be defined according to (5) from $m_k$ samples collected by $k$. Then, there exists some constants $\beta_{k,n}$ (depending on $n, k, p$) and $\xi_{k,n,F}$ (additionally depending on $F$) such that, with probability $1 - \delta$,*

$$
|\widehat{r}_k(n) - r(n)| \leq \beta_{k,n}\sqrt{\frac{\log\left(\frac{2\lceil n\sqrt{m_k}\rceil}{\delta}\right)}{m_k}} + n \times \xi_{k,n,F}\left(\frac{\log\left(\frac{2\lceil n\sqrt{m_k}\rceil}{\delta}\right)}{m_k}\right)^{\frac{n+p-1}{k+p}} . \tag{7}
$$

*Furthermore, the constants admit universal upper bounds for any $n, k, p, F$. For instance if $m_k \geq 4$ it holds that $\beta_{k,n} \leq 33$ and $\gamma_{k,n,F} \leq 100$.*

*Proof.* We build the proof from the Riemann sum approximation of the reward presented in Lemma 4 and the Chernoff bounds presented in Lemma 5. Defining some parameter $D \in \mathbb{N}$ that will be fixed later, we use the first result to consider the following approximation of the empirical reward estimate $\widehat{r}_k(n)$ by

$$
\widetilde{r}_k(n) = \frac{1}{D}\sum_{s=0}^{D-1}\left(\widehat{F}_{k+p,m_k}\left(\frac{s}{D}\right)^{\frac{n+p-1}{k+p}} - \widehat{F}_{k+p,m_k}\left(\frac{s}{D}\right)^{\frac{n+p}{k+p}}\right) .
$$

Thanks to Lemma 4, we know that $\widehat{r}_k(n) \in \left[\widetilde{r}_k(n) - \frac{n}{D}, \widetilde{r}_k(n) + \frac{n}{D}\right]$. For the rest of proof, we introduce the notation $F_s^j = F\left(\frac{s}{D}\right)^j$ for any $j \in [N]$, $\widehat{F}_s^{k+p} = \widehat{F}_{k+p,m_k}\left(\frac{s}{D}\right)$, and $\widehat{F}_{s,k,n} =$

$\widehat{F}_{k+p,m_k}\left(\frac{s}{D}\right)^{\frac{n+p}{k+p}}$, so that

$$\widetilde{r}_k(n) := \frac{1}{D}\sum_{s=0}^{D-1}\left(\widehat{F}_{s,k,n-1} - \widehat{F}_{s,k,n}\right) \ ,$$

that we want to relate with

$$\widetilde{r}(n) := \frac{1}{D}\sum_{s=0}^{D-1}\left(F_s^{n+p-1} - F_s^{n+p}\right) \ .$$

We use that each variable $\widehat{F}_{s,k,n}$ can be expressed as the expectation of $m_k$ i.i.d. Bernoulli random variables of expectation $F_s^{k+p}$, since $\widehat{F}_{k+p,m_k} = \frac{1}{m_k}\sum_{j=1}^{m_k}\mathbb{1}\{X_{k,j} \leq x\}$. Hence, we can use the confidence intervals providing by Lemma 5, according to the value of $F_s^{k+p}$. We define two critical values, corresponding to the switch between the different intervals $I_0, I_1, I_2$ in the lemma, and their closest upper point in the discretization grid. The first is

$$x_{0,k} = F^{-1}\left(\left(\frac{\delta}{m_k}\right)^{\frac{1}{k+p}}\right), \quad \text{and} \quad s_{0,k} = \lceil Dx_{0,k}\rceil \ ,$$

and we recall that below $x_{0,k}$ it holds that $\widehat{F}_{k,i,n-1} = 0$ with probability larger than $1 - \delta$. Then, we define

$$x_{1,k} := F^{-1}\left(1 \wedge \left(4\frac{\log\left(\frac{2}{\delta}\right)}{m_k}\right)^{\frac{1}{k+p}}\right), \quad \text{and} \quad s_{1,k} := \lceil Dx_{1,k}\rceil \ .$$

We remark that we use a multiplicative factor $4$ inside of $F^{-1}$, while Lemma 5 might suggest to use 3. We do that for technical reasons, that we will motivate at one stage of the proof. These terms depend both on the sample size $m_k$ and the confidence level $\delta$, but we omit them in the notation for simplicity. Then, for fixed values of these constants we decompose the estimator between the intervals $I_0 = \{s_{1,k}, \ldots, D-1\}$, $I_1 = \{s_{0,k}, \ldots, s_{1,k}-1\}$, and $I_2 = \{0, \ldots, s_{0,k}-1\}$. Note that for the second interval to be non-empty it must hold that $s_{0,k} \leq s_{1,k} - 1$, that we assume in the following, otherwise we can just remove this interval from the analysis.

For $s \geq s_{1,k}$, Lemma 5 guarantees that with probability larger than $1 - \delta$ it holds that

$$\widehat{F}_s^{k+p} \in \left[(1 - \gamma_s)F_s^{k+p}, (1 + \gamma_s)F_s^{k+p}\right] \quad \text{for } \gamma_s = F_s^{-\frac{k+p}{2}}\sqrt{3\frac{\log\left(\frac{2}{\delta}\right)}{m_k}} \ ,$$

while for $k \leq s_{1,k} - 1$ we can use one of the two other bounds provided in the lemma. Using a union bound, all the confidence intervals hold simultaneously for the points in the sum and in $x_{1,k}$ with probability larger than $1 - (D+1)\delta$, which defines a "good" event

$$\mathcal{G} = \left\{\forall k \in I_2, \widehat{F}_s^{k+p} = 0, \ \ \forall k \in I_1, \ \widehat{F}_s^{k+p} \leq 8\frac{\log\left(\frac{2}{\delta}\right)}{m_k}, \right.$$

$$\left. \forall k \in I_0, \widehat{F}_s^{k+p} \in \left[(1 - \gamma_s)F_s^{k+p}, (1 + \gamma_s)F_s^{k+p}\right]\right\} \ . \tag{9}$$

For the rest of the analysis, we assume that $\mathcal{G}$ holds. In particular, in that context there exists $D - s_{1,k}$ constants $(z_s)_{s \in \{s_{1,k},\ldots,D-1\}}$ such that $\forall s \in I_0, \widehat{F}_s^{k+p} = (1 + z_s)F_s^{k+p}$ and $z_s \in [-\gamma_s, \gamma_s]$. We now upper and lower bound $\widehat{r}_k(n)$ using these constants, first writing that under $\mathcal{G}$ it holds that

$$\widetilde{r}_k(n) = \frac{1}{D}\sum_{s=0}^{D-1}\left(\widehat{F}_{s,k,n-1} - \widehat{F}_{s,k,n}\right)$$

$$= \frac{1}{D}\sum_{s=s_{1,k}}^{D-1}\left(\widehat{F}_{s,k,n-1} - \widehat{F}_{s,k,n}\right) + \frac{1}{D}\sum_{s=0}^{s_{1,k}-1}\left(\widehat{F}_{s,k,n-1} - \widehat{F}_{s,k,n}\right)$$

$$= \frac{1}{D}\sum_{s=s_{1,k}}^{D-1}\left((1 + z_s)^{\frac{n+p-1}{k+p}}F_s^{n+p-1} - (1 + z_s)^{\frac{n+p}{k+p}}F_s^{n+p}\right) + \frac{1}{D}\sum_{s=s_{0,k}}^{s_{1,k}-1}\left(\widehat{F}_{s,k,n-1} - \widehat{F}_{s,k,n}\right) \ ,$$

where we used that all the terms are zero for indices smaller than $s_{0,k}$. We can thus express $\widehat{r}_k(n)$ as follows,

$$\widehat{r}_k(n) = \widetilde{r}(n) + n\mathcal{E}_0 + n\mathcal{E}_1 ,$$

with

$$\mathcal{E}_0 := \frac{1}{D} \sum_{s=s_{1,k}}^{D-1} ((1+z_s)^{\frac{n+p-1}{k+p}} - 1)F_s^{n+p-1} - \frac{1}{D} \sum_{s=s_{1,k}}^{D-1} ((1+z_s)^{\frac{n+p}{k+p}} - 1)F_s^{n+p} , \quad \text{and}$$

$$\mathcal{E}_1 := \frac{1}{D} \sum_{s=s_{0,k}}^{s_{1,k}-1} \left( \widehat{F}_{s,k,n-1} - \widehat{F}_{s,k,n} \right) - \sum_{s=0}^{s_{1,k}-1} \left( F_s^{n+p-1} - F_s^{n+p} \right) ,$$

so we can upper and lower bound $\widehat{r}_i(n)$ by upper and lower bounding $\mathcal{E}_0$ and $\mathcal{E}_1$ separately.

**Bounding the individual terms of $\mathcal{E}_0$**   For any $k \geq s_{1,k}$ we consider the term

$$\mathcal{E}_{0,s} := ((1+z_s)^{\frac{n+p-1}{k+p}} - 1)F_s^{n+p-1} - ((1+z_s)^{\frac{n+p}{k+p}} - 1)F_s^{n+p} .$$

We first re-arrange it in a more convenient way, remarking that

$$F_s^{n+p-1} = F_s^{n+p-1}(F_s + (1 - F_s)) = F_s^{n+p} + F_s^{n+p-1}(1 - F_s).$$

Using this result, we obtain that

$$\begin{aligned}
\mathcal{E}_{0,s} &:= ((1+z_s)^{\frac{n+p-1}{k+p}} - 1)F_s^{n+p-1} - ((1+z_s)^{\frac{n+p}{k+p}} - 1)F_s^{n+p} \\
&= ((1+z_s)^{\frac{n+p-1}{k+p}} - 1)F_s^{n+p-1}(F_s + (1 - F_s)) - ((1+z_s)^{\frac{n+p}{k+p}} - 1)F_s^{n+p} \\
&= F_s^{n+p}((1+z_s)^{\frac{n+p-1}{k+p}} - 1) - (1+z_s)^{\frac{n+p}{k+p}} + 1) + F_s^{n+p-1}(1 - F_s)((1+z_s)^{\frac{n+p-1}{k+p}} - 1) ,
\end{aligned}$$

which simplifies to

$$\mathcal{E}_{0,s} = \underbrace{F_s^{n+p}((1+z_s)^{\frac{n+p-1}{k+p}} - (1+z_s)^{\frac{n+p}{k+p}})}_{\mathcal{E}_{0,s}^-} + \underbrace{F_s^{n+p-1}(1 - F_s)((1+z_s)^{\frac{n+p-1}{k+p}} - 1)}_{\mathcal{E}_{0,s}^+} \tag{10}$$

We remark that these two terms have opposite sign, $\mathcal{E}_{0,s}^+$ having the same sign as $z_s$. We first upper bound $\mathcal{E}_{0,s}$, starting with the case $z_s > 0$, for which it holds that

$$z_s \geq 0 \Rightarrow \mathcal{E}_{0,s} \leq \mathcal{E}_{0,s}^+ \leq \underbrace{((1+\gamma_s)^{\frac{n+p-1}{k+p}} - 1)}_{c_s} F_s^{n+p-1}(1 - F_s) .$$

The constant $c_k$ is explicit from the definition of $\gamma_s$, and the bound holds for any $i \in [N+p]$ without restriction. However, if we only consider the case $\frac{n+p-1}{k+p} \leq 2$ then we can further write that

$$c_s \leq (1+\gamma_s)^2 - 1 \leq 2\gamma_s + \gamma_s^2 \leq 3\gamma_s ,$$

since $\gamma_s \leq 1$ for the values of $s$ considered. Then, for $z_s \leq 0$ we use that

$$\begin{aligned}
z_s \leq 0 \Rightarrow \mathcal{E}_{0,s} \leq \mathcal{E}_{0,s}^- &\leq F_s^{n+p}((1+z_s)^{\frac{n+p-1}{k+p}} - (1+z_s)^{\frac{n+p}{k+p}}) \\
&= F_s^{n+p}(1+z_s)^{\frac{n+p-1}{k+p}} \left(1 - (1+z_s)^{\frac{1}{k+p}}\right) .
\end{aligned}$$

Using the notation $y_s = -z_s$ for convenience, we upper bound the last multiplicative term as follows,

$$\begin{aligned}
(1 - y_s)^{\frac{1}{k+p}} = e^{\frac{\log(1-y_s)}{k+p}} &\geq 1 + \frac{\log(1 - y_s)}{k + p} \\
&= 1 - \frac{1}{k + p} \log \left(1 + \frac{y_s}{1 - y_s}\right) \\
&\geq 1 - \frac{1}{k + p} \times \frac{y_s}{1 - y_s} ,
\end{aligned}$$

which leads to

$$y_s := -z_s \geq 0 \Rightarrow \mathcal{E}_{0,s} \leq \mathcal{E}_{0,s}^- \leq F_s^{n+p}(1-y_s)^{\frac{n+p-1}{k+p}} \frac{y_s}{(k+p)(1-y_s)}$$

$$= F_s^{n+p}(1-y_s)^{\frac{n+p-1}{k+p}-1} \frac{y_s}{k+p} .$$

We now remark that when $k+p \leq n+p-1$ then the bound simply becomes $\mathcal{E}_{0,s}^- \leq F_s^{n+p} \times \frac{\gamma_s}{k+p}$. However, when $k+p > n+p-1$ the upper bound is diverging when $y_s$ gets close to 1. Since the upper bound is increasing in $\gamma_s$, and using that $n+p-1 \geq \frac{2}{3}(k+p)$ we obtain that $\mathcal{E}_{0,s}^- \leq F_s^{n+p} \frac{\gamma_s}{i(1-\gamma_s)^{\frac{1}{3}}}$.

This is the motivation for calibrating the threshold $s_{1,k}$ so that $k \geq s_{1,k} \Rightarrow (1-\gamma_s)^{\frac{1}{3}} \geq \frac{1}{2}$, which is done by tuning $s_{1,k}$ so that $\gamma_{s_{1,k}} \leq \frac{7}{8}$ (hence the multiplicative 4 inside of $F^{-1}$).

We thus obtain that

$$z_s \leq 0, k \geq s_{1,k} \Rightarrow \mathcal{E}_{0,s} \leq 2F_s^{n+p} \frac{\gamma_s}{k+p} ,$$

which finally leads to

$$\mathcal{E}_{0,s} \leq \underbrace{3\gamma_s F_s^{n+p-1}(1-F_s)}_{\text{if } z_s \geq 0} \vee \underbrace{2 F_s^{n+p} \frac{\gamma_s}{k+p}}_{\text{if } z_s \leq 0} . \tag{11}$$

We now proceed to lower bound $\mathcal{E}_{0,s}$, using again Equation(10). The proof is similar to the proof of the upper bound, for the case $z_s \geq 0$ we can write that

$$z_s \geq 0 \Rightarrow -\mathcal{E}_{0,s} \leq -\mathcal{E}_{0,s}^- \leq F_s^{n+p}(1+z_s)^{\frac{n+p-1}{k+p}}\left((1+z_s)^{\frac{1}{k+p}} - 1\right)$$

$$\leq F^{n+p}(1+\gamma_s)^{\frac{n+p-1}{k+p}} \frac{\gamma_s}{k+p}$$

$$\leq 2^{\frac{n+p-1}{k+p}} F^{n+p} \frac{\gamma_s}{k+p} ,$$

using the concavity of $x \mapsto (1+x)^{\frac{1}{k+p}}$ and that $\gamma_s \leq 1$. Then, for the case $z_s \leq 0$ we use that

$$z_s \leq 0 \Rightarrow -\mathcal{E}_{0,s} \leq -\mathcal{E}_{0,s}^+ \leq F_s^{n+p-1}(1-F_s)\left(1-(1+z_s)^{\frac{n+p-1}{k+p}}\right)$$

$$\leq F_s^{n+p-1}(1-F_s)\left(1-(1-\gamma_s)^{\frac{n+p-1}{k+p}}\right).$$

If $\frac{n+p-1}{k+p} \leq 2$ we furthermore obtain that $(1-\gamma_s)^{\frac{n+p-1}{k+p}} \geq (1-\gamma_s)^2 \geq 1-2\gamma_s$, so

$$z_s \leq 0 \Rightarrow -\mathcal{E}_{0,s}^+ \leq 2\gamma_s F_s^{n+p-1}(1-F_s) .$$

Combining these results, we can lower bound $\mathcal{E}_{0,s}$ as follows,

$$\mathcal{E}_{0,s} \geq -\left\{\frac{2^{\frac{n+p-1}{k+p}}\gamma_s}{k+p}F_s^{n+p} \vee 2\gamma_s F_s^{n+p-1}(1-F_s)\right\} , \tag{12}$$

where the terms involved in this lower bound are analogous to the terms used in the upper bound up to some multiplicative constants.

**Summary: bounds on $\mathcal{E}_0$**   We start with the lower bound. Using Equation (12), we obtain that

$$n\mathcal{E}_0 \geq -\frac{n}{D}\sum_{s=s_{1,k}}^{D-1}\left\{\frac{2^{\frac{n+p-1}{k+p}}\gamma_s}{k+p}F_s^{n+p} \vee 2\gamma_s F_s^{n+p-1}(1-F_s)\right\}$$

$$\geq -\sqrt{\frac{3\log\left(\frac{2}{\delta}\right)}{m_k}} \times \left\{\frac{n}{D}\sum_{s=s_{1,k}}^{D-1} 2^{\frac{n+p-1}{k+p}}\frac{F_s^{n+p-\frac{k+p}{2}}}{k+p} + \frac{n}{D}\sum_{s=s_{1,k}}^{D-1} 2F_s^{n+p-1-\frac{k+p}{2}}(1-F_s)\right\}$$

The first sum can be trivially upper bounded by $2^{\frac{n+p-1}{k+p}}\frac{n}{k+p}$, which cannot be refined without more restrictive assumptions on $F$. For the second term, we use that $k+p \leq \frac{3}{2}(n+p)$ to exhibit the reward function associated with a number of players $n' := n - \frac{n(k+p)}{2(n+p)} \geq \frac{n}{4}$ and a competition of size $p' := p - \frac{p(k+p)}{2(n+p)} \geq \frac{p}{4}$.

$$
\begin{aligned}
\frac{n}{D}\sum_{s=s_{1,k}}^{D-1} F_s^{n+p-1-\frac{k+p}{2}}(1-F_s) &= n \times \frac{1}{D}\sum_{s=s_{1,k}}^{D-1} F_s^{n'+p'-1}(1-F_s) \\
&\leq n \times \int_0^1 F(x)^{\frac{n+p-1}{4}}(1-F(x))\mathrm{d}x + \frac{n}{D} \\
&= \frac{n}{n'} \times n' \int_0^1 F(x)^{n'+p'-1}(1-F(x))\mathrm{d}x + \frac{n}{D} \\
&\leq \frac{n}{n'+p'-1} + \frac{n}{D} = \frac{n}{n+p-\frac{k+p}{2}} + \frac{n}{D} \ ,
\end{aligned}
$$

where we used that the reward is smaller than the probability that the coalition wins the auction, which is easily generalized even if $i/2$ is not integer.

We thus conclude the proof of the lower bound by writing that

$$
n\mathcal{E}_0 \geq -4\left\{\left(\frac{n}{2(n+p)-(k+p)} + \frac{n}{2D}\right) + 2^{\frac{n+p-1}{k+p}-2} \times \frac{n}{k+p}\right\} \times \sqrt{\frac{3\log\left(\frac{2}{\delta}\right)}{m_k}} \ , \tag{13}
$$

where the worst scaling for the left-hand term in the maximum is attained in $k+p = 3\frac{n+p}{2}$ and provide $2\frac{n}{n+p}$, while for the right-hand term it is achieved in $k+p = \frac{n+p-1}{2}$ and provides $2\frac{n}{n+p-1}$.

As we already discussed, the upper bound can be expressed very similarly, remarking that the bound involving the terms $\gamma_s/i$ have to be divided by two, and the other term have to be multiplied by $3/2$. We hence directly obtain that

$$
n\mathcal{E}_0 \leq 6\left\{\left(\frac{n}{2(n+p)-(k+p)} + \frac{n}{2D}\right) + \frac{n}{3(k+p)}\right\} \times \sqrt{\frac{3\log\left(\frac{2}{\delta}\right)}{m_k}} \ . \tag{14}
$$

**Bounds on $\mathcal{E}_1$** We start by upper bounding the second sum by $0$. Under $\mathcal{G}$ we hence obtain that

$$
\begin{aligned}
n\mathcal{E}_1 &\leq \frac{n}{D}\sum_{k\in I_1}(\widehat{F}_s^{k+p})^{\frac{n+p-1}{k+p}} \\
&\leq \frac{n}{D}\sum_{k\in I_1}\left(8\frac{\log(2/\delta)}{m_k}\right)^{\frac{n+p-1}{k+p}} \\
&\leq n\left(x_{1,k} - x_{0,k} + \frac{1}{D}\right)\left(8\frac{\log(2/\delta)}{m_k}\right)^{\frac{n+p-1}{k+p}} \ ,
\end{aligned}
$$

which has a worst possible power of $2/3$ when $m_k \geq 8\log(2/\delta)$, corresponding to $k+p = \frac{3}{2}(n+p-1)$. Replacing $x_{1,k}$ by its expression, we further obtain that

$$
n\mathcal{E}_1 \leq n\left(8\frac{\log(2/\delta)}{m_k}\right)^{\frac{n+p-1}{k+p}} \times \left\{F^{-1}\left(1 \wedge \left(\frac{4\log\left(\frac{2}{\delta}\right)}{m_k}\right)^{\frac{1}{k+p}}\right) - F^{-1}\left(\left(\frac{\delta}{m_k}\right)^{\frac{1}{k+p}}\right) + \frac{1}{D}\right\} \ . \tag{15}
$$

For the lower bound on $n\mathcal{E}_1^1$, we apply the exact same steps, remarking that the constant $8$ at the very first step can be replaced by $4$, since we now use the exact value of $F^{k+p}$ in the upper bound. Furthermore, we have to remove the term $x_{0,k}$. We finally obtain

$$
n\mathcal{E}_1 \geq -n\left(4\frac{\log(2/\delta)}{m_k}\right)^{\frac{n+p-1}{k+p}} \times \left\{F^{-1}\left(1 \wedge \left(\frac{4\log\left(\frac{2}{\delta}\right)}{m_k}\right)^{\frac{1}{k+p}}\right) + \frac{1}{D}\right\} \ . \tag{16}
$$

**Summary: bounds on** $\widehat{r}_k(n)$    We conclude this proof by summarizing the results, and exhibiting the constants introduced in the theorem. First, by combining (13) and (16) we obtain the following lower bound,

$$\widehat{r}_k(n) \geq r(n) - \frac{n}{D} - n \left(4\frac{\log(2/\delta)}{m_k}\right)^{\frac{n+p-1}{k+p}} \times \left\{ F^{-1}\left(1 \wedge \left(\frac{4\log\left(\frac{2}{\delta}\right)}{m_k}\right)^{\frac{1}{k+p}}\right) + \frac{1}{D} \right\}$$

$$- 4 \left\{ \left(\frac{n}{2(n+p) - (k+p)} + \frac{n}{2D}\right) + 2^{\frac{n+p-1}{k+p}-2} \times \frac{n}{k+p} \right\} \times \sqrt{\frac{3\log\left(\frac{2}{\delta}\right)}{m_k}} \ .$$

Then, by combining (14) and (15) we obtain the following upper bound,

$$\widehat{r}_k(n) \leq r(n) + \frac{n}{D} + 6 \left\{ \left(\frac{n}{2(n+p) - (k+p)} + \frac{n}{2D}\right) + \frac{n}{3(k+p)} \right\} \times \sqrt{\frac{3\log\left(\frac{2}{\delta}\right)}{m_k}}$$

$$+ n \left(8\frac{\log(2/\delta)}{m_k}\right)^{\frac{n+p-1}{k+p}} \times \left\{ F^{-1}\left(1 \wedge \left(\frac{4\log\left(\frac{2}{\delta}\right)}{m_k}\right)^{\frac{1}{k+p}}\right) - F^{-1}\left(\left(\frac{\delta}{m_k}\right)^{\frac{1}{k+p}}\right) + \frac{1}{D} \right\} \ .$$

As a final step, we recall that in this proof $1 - \delta$ is the confidence level of the point estimate in each of the $D$ points $(x_s)_{s \in \{0,\ldots,D-1\}}$ and $x_{1,k}$. Hence, to obtain a confidence $1 - \delta$ on the full estimate $\widehat{r}_k(n)$ we need to multiply $\delta$ by $(D+1)$ in the bounds presented above. As a final step, we choose $D + 1 = \lceil n\sqrt{m} \rceil$, so that the term $\frac{n}{D} \leq \frac{1}{\sqrt{m_k}}$ becomes a second order term in the bounds.

After replacing $\delta$ and $D$ by the appropriate values, we hence obtain that

$$\widehat{r}_k(n) \geq r(n) - \frac{1}{\sqrt{m_k}} - \beta_{k,n}^- \sqrt{\frac{\log\left(\frac{2\lceil n\sqrt{m_k}\rceil}{\delta}\right)}{m_k}} - n \times \xi_{k,n,F}^- \left(\frac{\log\left(\frac{2\lceil n\sqrt{m_k}\rceil}{\delta}\right)}{m_k}\right)^{\frac{n+p-1}{k+p}} . \quad (17)$$

with

$$\beta_{k,n}^- = 4\sqrt{3} \left\{ \left(\frac{n}{2(n+p) - (k+p)} + \frac{n}{2(\lceil n\sqrt{m_k}\rceil - 1)}\right) + 2^{\frac{n+p-1}{k+p}-2} \times \frac{n}{k+p} \right\} , \text{ and}$$

$$\xi_{k,n,F}^- = 4^{\frac{n+p-1}{k+p}} \left\{ F^{-1}\left(1 \wedge \left(\frac{4\log\left(\frac{2}{\delta}\right)}{m_k}\right)^{\frac{1}{k+p}}\right) + \frac{1}{\lceil n\sqrt{m_k}\rceil - 1} \right\} . \quad (18)$$

Symmetrically, we obtain that

$$\widehat{r}_k(n) \leq r(n) + \frac{1}{\sqrt{m_k}} + \beta_{k,n}^+ \sqrt{\frac{\log\left(\frac{2\lceil n\sqrt{m_k}\rceil}{\delta}\right)}{m_k}} + n \times \xi_{k,n,F}^+ \left(\frac{\log\left(\frac{2\lceil n\sqrt{m_k}\rceil}{\delta}\right)}{m_k}\right)^{\frac{n+p-1}{k+p}} . \quad (19)$$

with

$$\beta_{k,n}^+ = 6\sqrt{3} \left\{ \left(\frac{n}{2(n+p) - (k+p)} + \frac{n}{2(\lceil n\sqrt{m_k}\rceil - 1)}\right) + \frac{n}{3(k+p)} \right\} , \text{ and}$$

$$\xi_{k,n,F}^+ = 8^{\frac{n+p-1}{k+p}} \left\{ F^{-1}\left(1 \wedge \left(\frac{4\log\left(\frac{2}{\delta}\right)}{m_k}\right)^{\frac{1}{k+p}}\right) - F^{-1}\left(\left(\frac{\delta}{m_k}\right)^{\frac{1}{k+p}}\right) + \frac{1}{\lceil n\sqrt{m_k}\rceil - 1} \right\} .$$

$$(20)$$

Hence, we obtain the statement of (7) by choosing $\beta_{k,n} = \beta_{k,n}^- \vee \beta_{k,n}^-$ and $\xi_{k,n,F} = \xi_{k,n,F}^+ \vee \xi_{k,n,F}^-$. Furthermore, it is clear from their expression and the constraint $k + p \in \left[\frac{n+p}{2}, 3\frac{n+p}{2}\right]$ that these two constants are bounded by by absolute constants. Their expression provided in the theorem comes from choosing the worst admissible value of $k$ for each of their components. This concludes the proof of the theorem. $\qquad\square$

**Remark 2** (Improved constants for practical implementations). *We can further improve the constants of the bounds according to the position of $i$ with respect to $n + p - 1$.*

- *In Equation* (11) *(upper bound): the constant 3 can be improved to 1 if $i \geq n + p - 1$, while the constant 2 can be improved to $\frac{n+p-1}{k+p}$ if $i \leq n + p - 1$.*

- *In Equation* (12) *(lower bound): the constant 2 on the right-hand side can be improved to 1 if $n + p - 1 \leq i$.*

*These improved constants translate easily to the upper and lower bounds presented in* (13) *and* (14).

## B.3 Proof of Lemma 3

In this part, we prove the tighter confidence bounds, assuming that the quantile function of the value distribution is Lipschitz.

**Lemma 3** (Improved bound for Lipschitz quantile function). *Assume that $k \in \mathcal{V}(n)$ and $F^{-1}$ is $L$-Lipschitz, then there exists an absolute constant $\xi$ such that with probability $1 - \delta$ it holds that*

$$|\widehat{r}_k(n) - r(n)| \leq \beta_{k,n} \sqrt{\frac{\log\left(\frac{2\lceil n\sqrt{m_k}\rceil}{\delta}\right)}{m_k}} + \xi L \log\left(\frac{4\lceil n\sqrt{m_k}\rceil}{\delta}\right) \left(\frac{\log\left(\frac{2\lceil n\sqrt{m_k}\rceil}{\delta}\right)}{m_k}\right)^{\frac{n+p-1}{k+p}} \quad (8)$$

*Proof.* As the result indicates, the improvement comes from providing finer upper and lower bound on the term $n\mathcal{E}_1$ in the proof of Theorem 1. We can start the refined analysis from Equations (16) and (15).

Let us consider first the upper bound. In that case, the main ingredient comes from refining the upper bound of the difference $x_{1,k} - x_{0,k}$. To do that, we first provide an upper bound on $1 \wedge \left(\frac{4\log\left(\frac{2}{\delta}\right)}{m_k}\right)^{\frac{1}{k+p}} - \left(\frac{\delta}{m_k}\right)^{\frac{1}{k+p}}$. We recall that, as in the previous proof, this $\delta$ should be multiplied by $\lceil n\sqrt{m_k}\rceil$ at the end of the computation. We omit this term for now for simplicity. We consider a first case where the first term is equal to 1, which leads to

$$1 - \left(\frac{\delta}{m_k}\right)^{\frac{1}{k+p}} = 1 - e^{-\frac{1}{k+p}\log\left(\frac{m_k}{\delta}\right)}$$

$$\leq \frac{1}{k+p}\log\left(\frac{m_k}{\delta}\right)$$

$$\leq \frac{1}{k+p}\log\left(\frac{4\log(2/\delta)}{\delta}\right)$$

$$= \frac{1}{k+p} \times \left\{\log\left(\frac{4}{\delta}\right) + \log\log\left(\frac{2}{\delta}\right)\right\} .$$

For the alternative case we obtain a similar result,

$$\left(\frac{4\log\left(\frac{2}{\delta}\right)}{m_k}\right)^{\frac{1}{k+p}} - \left(\frac{\delta}{m_k}\right)^{\frac{1}{k+p}} = \left(\frac{4\log\left(\frac{2}{\delta}\right)}{m_k}\right)^{\frac{1}{k+p}} \left(1 - \left(\frac{\delta}{4\log(2/\delta)}\right)^{\frac{1}{k+p}}\right)$$

$$\leq \left(\frac{4\log\left(\frac{2}{\delta}\right)}{m_k}\right)^{\frac{1}{k+p}} \times \frac{1}{k+p} \times \left\{\log\left(\frac{4}{\delta}\right) + \log\log\left(\frac{2}{\delta}\right)\right\} ,$$

so the two upper bounds simplify to

$$\left\{ 1 \vee \left( \frac{4 \log \left( \frac{2}{\delta} \right)}{m_k} \right)^{\frac{1}{k+p}} \right\} \times \frac{1}{k+p} \times \left\{ \log \left( \frac{4}{\delta} \right) + \log \log \left( \frac{2}{\delta} \right) \right\}$$

Next, we use the assumption that $F^{-1}$ is $L$-Lipschitz to obtain that

$$n(x_{1,k} - x_{0,k}) := n \left( F^{-1} \left( 1 \wedge \left( \frac{4 \log \left( \frac{2}{\delta} \right)}{m_k} \right)^{\frac{1}{k+p}} \right) - F^{-1} \left( \left( \frac{\delta}{m_k} \right)^{\frac{1}{k+p}} \right) \right)$$

$$\leq \frac{Ln}{k+p} \times \left\{ \log \left( \frac{4}{\delta} \right) + \log \log \left( \frac{2}{\delta} \right) \right\}$$

$$\leq 4 \frac{Ln}{n+p} \times \log \left( \frac{4}{\delta} \right) \ .$$

By substituting $\delta$ by $\delta/(\lceil n\sqrt{m_k} \rceil)$ we obtain that the linear dependency in $n$ obtained with the previous analysis is refined to a $\log \left( \frac{4 \lceil n\sqrt{m_k} \rceil}{\delta} \right)$ for the upper bound, which matches the result at this point.

We now consider the lower bound, and work on refining the upper bound of the term $-n\mathcal{E}_1$ in the proof of Theorem 1. To do that, we consider a new intermediary point $x'_{0,k} = F^{-1} \left( \frac{\delta}{m \times n^{\frac{k+p}{n+p-1}}} \right)$. For the rest of the proof we get back to the discretized formulation of the error (with generic step $D^{-1}$), and upper bound

$$-n\mathcal{E}_1 \leq \underbrace{\frac{n}{D} \sum_{k \in I_1} F_s^{n+p-1}}_{n\mathcal{E}_1^0} + \underbrace{\frac{n}{D} \sum_{k \in I_2} F_s^{n+p-1}}_{n\mathcal{E}_1^1} \ .$$

We remark that we can use the upper bound provided for $n(x_{1,k}-x_{0,k})$ to upper bound $n\mathcal{E}_1^1$, obtaining the same result up to some multiplicative constants. Hence, it remains to upper bound $\mathcal{E}_1^0$, for which additional steps are needed. However, we will simply use the exact same trick as before: we consider another sub-interval $I'_2$, for which this time it holds that $k \in I'_2 \Rightarrow F_s^{k+p} \leq \frac{\delta}{m \times n^{\frac{k+p}{n+p-1}}}$. Then, we have that

$$n\mathcal{E}_1^0 = \frac{n}{D} \sum_{k \in I'_2} F_s^{n+p-1} + \frac{n}{D} \sum_{k \in I_2 \setminus I'_2} F_s^{n+p-1}$$

$$\leq \frac{n}{D} \sum_{k \in I'_2} \frac{1}{n} \left( \frac{\delta}{m_k} \right)^{\frac{n+p-1}{k+p}} + \frac{n}{D} \sum_{k \in I_2 \setminus I'_2} \left( \frac{\delta}{m_k} \right)^{\frac{n+p-1}{k+p}}$$

$$\leq \frac{|I'_2|}{D} \left( \frac{\delta}{m_k} \right)^{\frac{n+p-1}{k+p}} + n \times \frac{|I_2| - |I'_2|}{D} \left( \frac{\delta}{m_k} \right)^{\frac{n+p-1}{k+p}} \ .$$

Just as before, we show that $\frac{|I_0| - |I'_0|}{D}$ cannot be too large if the quantile function is Lipschitz. More precisely, we obtain that

$$\left( \frac{\delta}{m_k} \right)^{\frac{1}{k+p}} - \left( \frac{\delta}{m_k n^{\frac{k+p}{n+p-1}}} \right)^{\frac{1}{k+p}} = \left( \frac{\delta}{m_k} \right)^{\frac{1}{k+p}} \frac{\log(n)}{n+p-1} \ ,$$

so that we can finally write that

$$n\mathcal{E}_1^0 \leq \left( \frac{\delta}{m_k} \right)^{\frac{n+p-1}{k+p}} + n \times \left( \left( \frac{\delta}{m_k} \right)^{\frac{1}{k+p}} \times \frac{\log(n)}{n+p-1} \times L + \frac{1}{D} \right) \times \left( \frac{\delta}{m_k} \right)^{\frac{n+p-1}{k+p}}$$

$$\leq \left( \frac{\delta}{m_k} \right)^{\frac{n+p-1}{k+p}} + n \times \left( \frac{\log(n)}{n+p-1} \times L + \frac{1}{n\sqrt{m_k}} \right) \times \left( \frac{\delta}{m_k} \right)^{\frac{n+p-1}{k+p}} \ ,$$

which is sufficient to conclude, since the multiplicative constants to $m_k^{-\frac{n+p-1}{k+p}}$ are clearly dominated by $\log\left(\frac{4\lceil n\sqrt{m_k}\rceil}{\delta}\right)$. $\qquad\square$

## B.4 Empirical UCB and LCB

The UCB and LCB in Equation (17) and Equation (19) depend explicitly on the unknown $F$ via $\xi^+_{k,n,F}$ and $\xi^-_{k,n,F}$ and therefore cannot be used in the implementation of GG.

Below, we give empirical UCB ($\widehat{U}_k(n,\delta)$) and LCB ($\widehat{L}_k(n,\delta)$) by replacing $\xi^+_{k,n,F}$ and $\xi^-_{k,n,F}$ by empirical estimates $\widehat{\xi}^+_{k,n}$ and $\widehat{\xi}^-_{k,n}$:

$$\widehat{U}_k(n,\delta) = \widehat{r}_k(n) + \frac{1}{\sqrt{m_k}} + \beta^-_{k,n}\sqrt{\frac{\log\left(\frac{2\lceil n\sqrt{m_k}\rceil}{\delta}\right)}{m_k}} + n\times\widehat{\xi}^-_{k,n}\left(\frac{\log\left(\frac{2\lceil n\sqrt{m_k}\rceil}{\delta}\right)}{m_k}\right)^{\frac{n+p-1}{k+p}} \tag{21}$$

$$\widehat{L}_k(n,\delta) = \widehat{r}_k(n) - \frac{1}{\sqrt{m_k}} - \beta^+_{k,n}\sqrt{\frac{\log\left(\frac{2\lceil n\sqrt{m_k}\rceil}{\delta}\right)}{m_k}} - n\times\widehat{\xi}^+_{k,n}\left(\frac{\log\left(\frac{2\lceil n\sqrt{m_k}\rceil}{\delta}\right)}{m_k}\right)^{\frac{n+p-1}{k+p}} \tag{22}$$

where $\beta^-_{k,n}$ and $\beta^+_{k,n}$ are defined in Equation (18) and Equation (20), and

$$\widehat{\xi}^-_{k,n} = 4^{\frac{n+p-1}{k+p}}\left\{\frac{\hat{d}_{k+p}+1}{\lceil n\sqrt{m_k}\rceil-1}\right\},$$

$$\widehat{\xi}^+_{k,n} = 8^{\frac{n+p-1}{k+p}}\left\{\frac{\hat{d}_{k+p}+1}{\lceil n\sqrt{m_k}\rceil-1}\right\}$$

for

$$\hat{d}_{k+p} = \inf\{d\in\{0,\ldots,\lceil n\sqrt{m_k}\rceil-1\}, \hat{F}^{k+p}_d \geq 8\frac{\log(2\lceil n\sqrt{m_k}\rceil/\delta)}{m_k}\}$$

if the infimum exists and $\hat{d}_{k+p} = 1$ otherwise.

It is a corollary of Theorem 1 that $\widehat{U}_k(n)$ and $\widehat{L}_k(n)$ are indeed high probability upper and lower bounds of the true reward:

**Corollary 1** (Explicit upper and lower bounds)**.** *It holds that*

$$\mathbb{P}(\widehat{L}_k(n,\delta) \leq r(n) \leq \widehat{U}_k(n,\delta)) \geq 1-\delta$$

*Proof.* With $D = \lceil n\sqrt{m_k}\rceil$ and $x_{1,k} = F^{-1}\left(1\wedge\left(\frac{4\log\left(\frac{2\lceil n\sqrt{m_k}\rceil}{\delta}\right)}{m_k}\right)^{\frac{1}{k+p}}\right)$ the good event $\mathcal{G}$ defined in Equation (9) implies that with probability $1-\delta$ the following event $\mathcal{H}$ holds:

$$\mathcal{H} = \left\{\forall k \leq \lceil Dx_{1,k}\rceil, \hat{F}^{k+p}_s \leq 8\frac{\log\left(\frac{2\lceil n\sqrt{m_k}\rceil}{\delta}\right)}{m_k}\right\}.$$

Under $\mathcal{H}$, and since by definition $\hat{F}^{k+p}_{\hat{d}_{k+p}} \geq 8\frac{\log(2\lceil n\sqrt{m_k}\rceil/\delta)}{m_k}$, it holds that

$$\frac{\hat{d}_{k+p}}{D} \geq x_{1,k} = F^{-1}\left(1\wedge\left(\frac{4\log\left(\frac{2\lceil n\sqrt{m_k}\rceil}{\delta}\right)}{m_k}\right)^{\frac{1}{k+p}}\right)$$

This implies that $\widehat{\xi}^-_{k,n} \geq \xi^-_{k,n,F}$ and $\widehat{\xi}^+_{k,n} \geq \xi^+_{k,n,F}$.

Therefore, we can incorporate in the proof of Theorem 1 the fact that the good event $\mathcal{G}$ implies $\widehat{\xi}^-_{k,n} \geq \xi^-_{k,n,F}$ and $\widehat{\xi}^+_{k,n} \geq \xi^+_{k,n,F}$ and obtain the stated result. $\qquad\square$

# C  Regret analysis of Local-Greedy and Greedy-Grid

## C.1  Clarification on the feedback received by the algorithms

In this section, we consider the case where a feedback (in the form of a sample from a power of $F$) is gathered only when an auction is won. If this is not the case, the decision-maker only knows that the coalition lost the auction. Therefore, if at time $t$, $n_t$ agents are assigned to an auction and the auction is lost, it makes sense to continue assigning $n_t$ agents to the auction at time $t + 1$, in order to gather the information that the algorithm wanted to obtain. The meta algorithm called CoMAB for coalition multi-armed bandits described in Algorithm 3 implements this strategy.

---

**Algorithm 3** CoMAB

---

**Init:** $\mathcal{J}_0 = \varnothing, m = 1$
**Input:** Algo
**for** $t = 1 \ldots T$ **do**
    **if** $t > 1$ *and auction at* $t - 1$ *is not won* **then**
        Play $n_t = n_{t-1}$ ;                  ▷ `Play same arm until an auction is won`
    **else**
        Play $n_t = \text{Algo}(\mathcal{J}_{m-1})$ ;    ▷ `When an auction is won play as prescribed by input`
        `Algo`
    **if** *Auction is won* **then**
        Observe $w_{n_t}$ a sample from $F^{n_t+p}$
        Set $\mathcal{J}_m = \mathcal{J}_{m-1} \cup \{(w_{n_t}, n_t, m)\}$ ;        ▷ `Record feedback obtained when winning`
        Update $m = m + 1$

---

Local-Greedy and Greedy-Grid are then defined as CoMAB applied on $\pi_{\text{LG}}$ and $\pi_{\text{GG}}$ for local greedy and greedy-grid respectively. These policies associate a play $n_m$ to an history $\mathcal{J}_{m-1}$. In Algorithm 1 and Algorithm 2, the policies are called sequentially $T$ times and feedback is observed after each request. This gives an implicit definition of the policies.

Lemma 6 expresses the regret of CoMAB in function of the behavior of any policy $\pi$ when feedback is observed after each request.

**Lemma 6** (Regret of CoMAB). *Consider a policy $\pi$ that associate to every $\mathcal{J}_{m-1}$ a play $n_m^\pi$. Define $\mathcal{J}_0 = \varnothing$ and $\mathcal{J}_m = \mathcal{J}_{m-1} \cup \{w_{n_m^\pi}, n_m^\pi, m\}$ where $w_{n_m^\pi}$ is a sample from a distribution with c.d.f $F^{n_m^\pi+p}$ and $n_m^\pi = \pi(\mathcal{J}_{m-1})$. Consider $m_n^\pi(m)$ the number of times $\pi$ returns $n$ after $m$ calls of $\pi$.*

*After $T$ iterations, CoMAB based on $\pi$ has regret:*

$$\mathcal{R}_T \leq \sum_{n=1}^{N} \mathbb{E}[m_n^\pi(T)] \frac{p+n}{n}(r(n^*) - r(n))$$

*Proof.* Call $n_t$ the play chosen by CoMAB at time $t$,

$$\eta_t = \mathbb{1}\{\text{The auction is won at time } t\},$$

$$m_n(t) = |\{\rho \leq t, n_\rho = n \text{ and } \eta_\rho = 1\}|$$

the number of times that $n$ is played and the auction is won up to time $t$ and

$$Z_{n,m}(t) = |\{\rho \leq t, n_\rho = n \text{ and } m \leq m_n(\rho) < m + 1\}|$$

the number of times that $n$ has been played between the $m$-th time $n$ won an auction and the $m + 1$-th time. Note that $m_n(t) \leq m_n^\pi(t)$ since at time $t$, $\pi$ has been called at most $t$ times.

The regret of CoMAB satisfies:

$$\mathcal{R}_T = \sum_{t=1}^{T} \mathbb{E}[r(n^*) - r(n_t)]$$

$$= \sum_{n=1}^{N} \mathbb{E}\left[\sum_{t=1}^{T} \mathbb{1}\{n_t = n\}\right](r(n^*) - r(n))$$

$$= \sum_{n=1}^{N} \mathbb{E}\left[\sum_{m=1}^{m_n(T)} Z_{n,m}(T)\right](r(n^*) - r(n))$$

$$= \sum_{n=1}^{N} \mathbb{E}\left[\sum_{m=1}^{m_n(T)} Z_{n,m}(T)\right](r(n^*) - r(n))$$

$$\leq \sum_{n=1}^{N} \mathbb{E}[m_n^{\pi}(T)]\frac{p+n}{n}(r(n^*) - r(n))$$

where in the second to last inequality, we used the independence between $n_t = n$ and $Z_{n,m_t(n)}(T)$ as $n_t = n$ depends only on the history at times $t < m_t(n)$ while $Z_{n,m_t(n)}(T)$ depends only on times $t \geq m_t(n)$.

$\square$

## C.2 Auxiliary result

Before proving the theorems, we present an auxiliary result from [6] that we to derive upper bounds that can be recovered explicit in the proof of the theorems. Since the proof is simple, we recall it for completeness.

**Lemma 7** (Lemma 4 from [6]). *For any $\zeta \geq 1$, the mapping*

$$f_\zeta : x \in [(\zeta+2)^\zeta \vee 3, \infty) \mapsto \sup\left\{t \in \mathbb{N} : \frac{t}{\log(t)^\zeta} \leq x\right\}$$

*satisfies*

$$f_\zeta(x) \leq (\zeta+2)^\zeta \times \log(x)^\zeta x.$$

*Proof.* We start by remarking that the function $g(x) = \frac{x}{\log(x)^\zeta}$ is strictly increasing for all $x \geq e^\zeta$. Now, consider a value $s = Ax\log(x)^\zeta$ for some $A > 0$, such that $s \geq 3 \vee e^\zeta$. By the monotonicity of $\frac{t}{(\log t)^\zeta}$, we have that

$$t > s \Rightarrow \frac{t}{(\log(t)^\zeta} > \frac{s}{(\log(s)^\zeta} = x \times \frac{A\log(x)^\zeta}{(\log(A) + \log(x) + \zeta\log(\log(x)))^\zeta} .$$

Then, for $x \geq A \geq 3$, it holds that $\log(A) + \log(x) + \zeta\log(\log(x)) \leq (\zeta+2)\log(x)$, so we can simply choose $A = (\zeta+2)^\zeta$ to obtain the result.

All that is left is to verify that for this choice, $s = (\zeta+2)^\zeta \times \log(x)^\zeta x \geq 3 \vee e^\zeta$, but this clearly holds for all $x \geq 3$ and $\zeta > 0$.

$\square$

## C.3 Proof of Theorem 2

**Theorem 2** (Regret bound for Local-Greedy). *Let $\Delta := \min_{n \in [N-1]} |r(n+1) - r(n)|$ (worst local gap). Under Assumption 2 and with $\alpha = (\log_{3/2} N + 1)^{-1}$, the regret of LG is upper bounded by a **problem-dependent constant**: there exists $(C_n)_{n \in [N] \setminus \{n^*\}}$, each satisfying $C_n = \widetilde{\mathcal{O}}_N\left(\frac{\Delta_n}{\Delta^2}\right)$, such that $\mathcal{R}_T \leq \sum_{n \in [N] \setminus n^*} C_n$.*

*Additionally, if the arm set forms a single estimation neighborhood, that is $\forall n \in [N] : \mathcal{V}(n) \supset [N]$, then each constant $C_n$ can be refined to $\widetilde{\mathcal{O}}_n\left(\Delta_n^{-1}\right)$, providing $\mathcal{R}_T = \widetilde{\mathcal{O}}(\sqrt{NT})$, which holds even when the reward function is not unimodal.*

*Proof.* First, we denote by $\widetilde{r}_t(n)$ the reward estimate used for arm $n$ at time $t$, and by $\widehat{r}_{k,t}(n)$ its value when the arm used to compute the estimate is fixed to $k \in \mathcal{V}(n)$. The proofs rely on concentration bounds on $\widetilde{r}_t(n)$ derived from Theorem 1, with a confidence level $\delta_t$ that will be fixed later. However, we will use this result with extra care given that the identity of the arm $k$ used to compute the estimate is a random variable, as well as its sample size $m_k(t)$. This issue is tackled with appropriate union bounds. Furthermore, in order to simplify the presentation we denote by $\mathcal{E}(m, \delta)$ the maximal diameter (as a function of $k$) of the confidence interval provided by Equation (7), defined by a number of plays $m$ of the arm used to estimate, and by a confidence level $\delta$. More precisely, with notation of Theorem 1, for any $(k, n) \in [N]^2$ we write that $|\widehat{r}_{k,t}(n) - r(n)| \leq \mathcal{E}(m_k(t), \delta_t)$ with probability at least $1 - \delta_t$. Furthermore, $\mathcal{E}(m_k(t), \delta_t)$ is increasing in $\delta_t$ and decreasing in $m_k(t)$. Finally, we use the notation $K = \lceil \log_{3/2}(N) \rceil$, so that $\alpha = \frac{1}{K+1}$.

We now prove the first statement of the theorem, by upper bounding the number of plays of each sub-optimal arm.

**Single neighborhood**  Consider any sub-optimal arm $n$. The main ingredient of the proof is to tackle the forced sampling by using that if $n$ is pulled at time $t$, then it is either pulled "on purpose" or due to forced sampling. However, it it forced sampled then it must have been selected "on purpose" by being the best empirical arm in the neighborhood at some previous point in time. We hence consider the following good event

$$\mathcal{G}_t = \{\forall s \in \{t - \lfloor \alpha t \rfloor, \dots, t\}, \ (\forall k \in \mathcal{V}(n) : m_k(s) \geq \alpha t), \ |\widehat{r}_{k,s}(n) - r(n)| \leq \mathcal{E}(m_k(s), \delta_s)\} \ .$$

Using Theorem 1, $\mathcal{G}_t$ holds with probability at least $1 - |\mathcal{V}(n)|t^2\delta_t$, where we used a crude union bound on the values of $s$, $k$ and $m_k(t)$ ($t$ could be replaced by $t - \lfloor \alpha \rfloor t \rfloor$). Using this result, we first upper bound the number of plays of $n$ up to horizon $T$ as follows,

$$\mathbb{E}\left[\sum_{t=1}^{T} \mathbb{1}\{n_t = n\}\right] \leq \mathbb{E}\left[\sum_{t=1}^{T} \mathbb{1}\{\exists s \in \{t - \lfloor \alpha t \rfloor, \dots, t\} : \widehat{r}_s(n) \geq \widehat{r}_s(n^\star)\}\right]$$

$$\leq \mathbb{E}\left[\sum_{t=1}^{T} \mathbb{1}\{\exists s \in \{t - \lfloor \alpha t \rfloor, \dots, t\} : \widehat{r}_s(n) \geq \widehat{r}_s(n^\star)\} \mathbb{1}\{\mathcal{G}_t\}\right] + \sum_{t=1}^{T} \mathbb{P}(\bar{\mathcal{G}}_t) \ .$$

As discussed above, the second term satisfies $\sum_{t=1}^{T} \mathbb{P}(\bar{\mathcal{G}}_t) \leq \sum_{t=1}^{+\infty} |\mathcal{V}(n)|t\delta_t$. In order to make it constant, we choose $\delta_t = \frac{1}{|\mathcal{V}(n)|t^4}$. For the first term, we use that $\forall s \in [\alpha t, t], \mathcal{E}(m_k(s), \delta_t) \leq \mathcal{E}(\alpha(1-\alpha)t, \delta_t)$ and that $n$ can be played only if the two confidence intervals overlap. Hence, we further obtain that

$$\mathbb{E}\left[\sum_{t=1}^{T} \mathbb{1}\{n_t = n\}\right] \leq \mathbb{E}\left[\sum_{t=1}^{T} \mathbb{1}\{2\mathcal{E}(\alpha(1-\alpha)t, \delta_t) \geq \Delta_n\}\right] + \sum_{t=1}^{+\infty} |\mathcal{V}(n)|t^2\delta_{t - \lfloor \alpha t \rfloor}$$

$$= \mathbb{E}\left[\sum_{t=1}^{T} \mathbb{1}\{2\mathcal{E}(\alpha(1-\alpha)t, \delta_t) \geq \Delta_n\}\right] + \sum_{t=1}^{+\infty} (1-\alpha)^{-4}t^{-2}$$

$$= t_{\alpha,n} + \frac{\pi^2}{6(1-\alpha)^4} \ ,$$

$$= t_{\alpha,n} + \frac{\pi^2}{6}\left(1 + \frac{1}{K}\right)^4 \ , \quad \text{since } \alpha = \frac{1}{K+1},$$

and for a deterministic constant $t_{\alpha,n}$ defined by

$$t_{\alpha,n} = \sup\{t \in \mathbb{N} : 2\mathcal{E}(\alpha(1-\alpha)t, \delta_t) \geq \Delta_n\} \ .$$

We recall that $K = \lceil \log_{3/2}(N) \rceil$ is used to simplify the notation. The last step consists in upper bounding the value of $t_{\alpha,n}$ explicitly according to $n$ and $\Delta_n$. Considering some $t \geq 4 \vee n + 1$, from

Theorem 1 we know that there exist universal constants $\beta$ and $\xi$, and we obtain that

$$\mathcal{E}(\alpha(1-\alpha)t, \delta_t) \leq \beta \sqrt{\frac{\log\left(\frac{2\lceil n\sqrt{m_k(s)}\rceil}{\delta_t}\right)}{m_k(s)}} + n \times \xi \left(\frac{\log\left(\frac{2\lceil n\sqrt{m_k(s)}\rceil}{\delta_t}\right)}{m_k(s)}\right)^{\frac{2}{3}}$$

$$\leq \beta \sqrt{\frac{\log\left(2t^4(n+1)\sqrt{t}|\mathcal{V}(n)|\right)}{\alpha(1-\alpha)t}} + n \times \xi \left(\frac{\log\left(2t^4(n+1)\sqrt{t}|\mathcal{V}(n)|\right)}{\alpha(1-\alpha)t}\right)^{\frac{2}{3}}$$

$$\leq \beta \sqrt{\frac{\log\left(t^5(n+1)^2\right)}{\alpha(1-\alpha)t}} + n \times \xi \left(\frac{\log\left(t^5(n+1)^2\right)}{\alpha(1-\alpha)t}\right)^{\frac{2}{3}}$$

$$\leq \beta \sqrt{\frac{7\log(t)}{\alpha(1-\alpha)t}} + n \times \xi \left(\frac{7\log(t)}{\alpha(1-\alpha)t}\right)^{\frac{2}{3}} .$$

Since $\alpha = \frac{1}{K+1}$ it holds that $\alpha(1-\alpha) = \frac{1}{K+1}\frac{K}{K+1} \geq \frac{1}{2(K+1)}$. We then get that for some universal constants $\beta'$ and $\xi'$, it first holds that:

$$\mathcal{E}(\alpha(1-\alpha)t, \delta_t) \leq \left(\beta'\sqrt{K+1} + n(K+1)^{\frac{2}{3}}\xi'\right)\sqrt{\frac{\log(t)}{t}}, \tag{23}$$

where we bounded $\left(\frac{\log(t)}{t}\right)^{\frac{2}{3}}$ by $\sqrt{\frac{\log(t)}{t}}$. Without this simplification, we also obtain that

$$\mathcal{E}(\alpha(1-\alpha)t, \delta_t) \leq (K+1)^{\frac{2}{3}}\left\{\beta' + n\left(\frac{\log(t)}{t}\right)^{\frac{1}{6}}\xi'\right\}\sqrt{\frac{\log(t)}{t}}. \tag{24}$$

The different scaling proposed in the theorem then come from using Lemma 7 on (23) and (24), taking the minimum between the two (since both bounds are valid simultaneously), and for the latter splitting cases depending on $\frac{t}{\log(t)} \leq n^6$ being satisfied or not (taking this time the maximum between the two cases).

We provide the right-hand term of the result using (23), applying Lemma 7 with $\zeta = 1$ and

$$x = \frac{4}{\Delta_n^2} \times \left(\beta'\sqrt{K+1} + n(K+1)^{\frac{2}{3}}\xi'\right)^2,$$

which leads to $t_{\alpha,n} \leq 3x\log(x)$. This provides the term $\mathcal{O}\left(\frac{n^2}{\Delta_n^2}\right)$ of the result, and constants in the logarithmic terms can be recovered explicitly by recovering the values of $\beta'$ and $\xi'$.

We then obtain the left-hand term of the result by considering (24). Lemma 7 first provides that $n\left(\frac{\log(t)}{t}\right)^{\frac{1}{6}} \leq 1$ for $t \geq 18n^6\log(n)$ (first bound). Still using Lemma 7, this simplification permits to use $\zeta = 1$ and the threshold

$$y = \frac{4}{\Delta_n^2} \times \left(\beta'\sqrt{K+1} + (K+1)^{\frac{2}{3}}\xi'\right)^2,$$

and an upper bound of $t_{\alpha,n} \leq 3y\log(y)$, but only if this term is larger than $18n^6\log(n)$. This provides the remaining terms of the bound, and again the logarithms can be easily recovered by computing $\beta'$ and $\xi'$.

This concludes the proof for the problem-dependent in the favorable case where all arms are neighbors, remarking that these upper bounds just have to be multiplied by $\Delta_n$ and summed over $n \in [N]$ to convert into the regret bound. Furthermore, the problem-independent guarantee can be derived from taking the minimum between the bound and $\Delta_n T$, remarking that the worst case is $\Delta_n = T^{-1/2}$ if we omit the logarithms.

**General case** We now provide the regret bound for the general case, where at least some arms do not include $[N]$ in their neighborhood. We recall that two main ingredients of Local-Greedy are (1) that the arm $n_t$ played in $t$ is the best empirical arm in the neighborhood of $\mathcal{V}(n_{t-1})$, according to the simple estimates computed with samples from $n_{t-1}$, and (2) that $n_t = n_{t-1}$ if $m_{n_{t-1}}(t) < \alpha t$ (forced sampling). Hence, similarly to the previous proof we use that $n_t$ is either pulled thanks to a "greedy play" or because of forced sampling. Furthermore, the two cases can be merged because forced sampling can only come after $n_t$ being pulled because of a greedy play in the recent rounds. More precisely, we use that

$$\{n_t = n\} \subset \{\exists s \in [t - \lfloor \alpha t \rfloor] : n_s = n, m_s(\ell_s) \geq \lceil \alpha s \rceil\} \ . \tag{25}$$

This argument is at the core of our analysis, but before going further we need to introduce the notion of *locally optimal plays*.

**Definition 2** (Locally optimal plays and optimal path). *Given a reference arm $\ell$, playing $n \in \mathcal{V}(\ell)$ is **locally optimal** if $n = \mathrm{argmax}_{k \in \mathcal{V}(\ell)} r(k)$. In that case, $n$ is the best neighbor of $\ell$, and we use the notation $n = v^+(n)$.*

*Furthermore, a sequence of successive locally optimal plays is an **optimal path** towards $n^\star$. By construction of $\mathcal{V}$, an optimal path contains at most $K := \mathcal{O}(\log(N))$ sub-optimal arms.*

The last fact presented in the definition is trivial: in the worst case the path start at one of the extremes of the interval $[N]$ and $n^\star$ is at the other. By design of $\mathcal{V}$ (Definition 1) we obtain that $n^\star$ is reached in $\lceil \log_{3/2}(N) \rceil$ steps at most. The rest of the proof is based on the idea that, when $t$ is large enough, the algorithm starts following an optimal path with high probability, so a sub-optimal arm can be played only if it is located on an optimal path from another sub-optimal arm to $n^\star$. We formalize it with the following result.

**Lemma 8** (Existence of a "recent" sub-optimal play). *For any time step $t \in [T]$ and arm $n \neq n^\star$, it holds that*

$$\{n_t = n\} \subset \mathcal{A}_t := \big\{\exists s \in \big[(1-\alpha)^K t, t\big], l \in [N], l' \in \mathcal{V}(l) : \widehat{r}_s(l) \leq \widehat{r}_s(l'),$$
$$m_s(l) \geq \alpha(1-\alpha)^K t \quad and \quad r(l) > r(l')\big\} \ ,$$

*Proof.* Starting from (25), we first use that either $n \neq v^+(\ell_s)$, and in that case playing $n$ is locally sub-optimal so this event belongs to $\mathcal{A}_t$, or $n = v^+(\ell_s)$. Let us now consider this second case: by definition of the leader, $\ell_s$ was played right before the sequence of forced plays of $n$ started, which must have happened at least as recently as $t - \lfloor \alpha t \rfloor - 1$. From that point, the recursion pattern is clear: $\ell_s$ must have been selected in the last $\lfloor \alpha(s-1) \rfloor$, by either being a locally sub-optimal play or not. The first case is included in $\mathcal{A}_t$, why the second requires to add another step in the analysis. Furthermore, the arm used to estimate $\ell_s$ was itself samples at least proportionally to $s$. Using that this can happen $K$ times, and that the worst number of steps to look into the past at each step is at most a fraction $(1 - \alpha)$ of the number in the previous step, we finally obtain that there have been a sub-optimal greedy play in the last $\big[(1 - \alpha)^K t, t\big]$ steps. $\qquad\square$

Before going further, we justify the tuning $\alpha = \frac{1}{K+1}$, by stating that it maximizes $\alpha(1 - \alpha)^K$ (used later in the proof). At this step, we can simplify the notation by remarking that

$$(1-\alpha)^K = \left(\frac{K}{K+1}\right)^K \geq e^{-1} \ ,$$

hence we replace $(1-\alpha)^K$ by $e^{-1}$ in the rest of the proof.

In words Lemma 8 states that, even if forced sampling slows down the ascension towards $n^\star$, since optimal paths contain at most $K$ sub-optimal arm then $n^\star$ is relatively fast to reach from any arm in $[N]$. Next, we use this result in the regret analysis by considering its occurrences with the following event,

$$\mathcal{H}_t = \big\{\forall s \in \big[e^{-1}t, t\big], \ \forall n \in [N], \ \forall k \in \mathcal{V}(n) : m_k(s) \geq \frac{e^{-1}}{1+K}t,$$
$$|\widehat{r}_{k,s}(n) - r(n)| \leq \mathcal{E}(m_k(s), \delta_s)\big\} \ .$$

Then, for any $t_K \in \mathbb{N}$ we can upper bound the number of plays of each sub-optimal arm $n \in [N]$ as follows,

$$\mathbb{E}\left[\sum_{t=1}^{T} \mathbb{1}\{n_t = n\}\right] \leq \mathbb{E}\left[\sum_{t \geq 1} \mathbb{1}\{\mathcal{A}_t\}\right]$$

$$\leq t_K + \mathbb{E}\left[\sum_{t \geq t_K} \mathbb{1}\{\mathcal{A}_t, \mathcal{H}_t\}\right] + \sum_{t \geq t_K} \mathbb{P}(\bar{\mathcal{H}}_t) \,,$$

with the slight abuse of notation that $\mathcal{E}$ is now defined with the coalition size $N$ and not the local value of $n$ considered. The rest of the proof is analogous to the simple case, where all arms are in a single neighborhood. We first choose $\delta_t = \frac{1}{N^2 t^4}$, and obtain that $\sum_{t \geq t_K} \mathbb{P}(\bar{\mathcal{H}}_t) \leq \sum_{t=1}^{+\infty} \frac{1}{e^{-4} t^2} \leq \frac{\pi^2}{6 e^{-4}}$.

Next, we tune $t_K$ large enough so that $\mathbb{E}\left[\sum_{t \geq t_K} \mathbb{1}\{\mathcal{A}_t, \mathcal{H}_t\}\right] = 0$. This can be done by choosing

$$t_K = \sup\left\{t \in \mathbb{N} : 2\mathcal{E}\left(\frac{e^{-1}}{1+K} t, \delta_t\right) \leq \Delta\right\} \,.$$

where $\Delta = \min_{n \in [N-1]}\{|r(n+1) - r(n)|\}$.

We then deduce the result by applying the exact same steps as for the upper bound of $t_{\alpha,n}$, by carefully replacing $\delta_t = \frac{1}{|\mathcal{V}(n)| t^4}$ by $\delta_t = \frac{1}{N^2 t^4}$, and $\alpha(1-\alpha)t$ by $\frac{e^{-1}}{1+K} t$. Lemma 7 then allows to easily obtain the desired scaling, and to make the upper bound explicit by substitution. Since $K$ is logarithmic in $N$, it is clear that it only contributes to the bound only by logarithmic factors. $\qquad\square$

## C.4 Proof of Theorem 3

**Theorem 3** (Regret upper bound for Greedy-Grid). *Suppose that* GG *is tuned with confidence level* $\delta_t = \frac{1}{N^2 t^3}$, *and* $\alpha = 1/4$. *Then, for any* $T \in \mathbb{N}$ *it holds that*

$$\mathcal{R}_T = \widetilde{\mathcal{O}}_N\left(\sum_{n \in \mathcal{B}^\star} \frac{1}{\Delta_n} + \sum_{n \in \mathcal{S}} \frac{\log(T)}{\Delta_n} \wedge \Delta_n \left(\frac{\mathbb{1}\{n < n^\star\}}{\Delta_{v_l(n^\star)}^2} + \frac{\mathbb{1}\{n > n^\star\}}{\Delta_{v_r(n^\star)}^2}\right)\right) \,.$$

*Additionally, it holds that* $\mathcal{R}_T = \widetilde{\mathcal{O}}\left(\sqrt{(K + |\mathcal{B}^\star|)T}\right)$, *for* $K = \lfloor \log_{3/2}(N) \rfloor$.

*Proof.* As the theorem suggests, we will use different arguments depending on the position of $n$ with respect to the grid and the optimal bin $\mathcal{B}^\star$. Before that, we introduce the crucial result of this proof: for each time $t$ large enough, thanks to the design of Greedy-Grid, all arms in optimal bin $\mathcal{B}^\star$ are estimated with a simple estimate computed with a *linear* number of samples in $t$.

Following the implementation of GG, at each time $t$ and for each arm $n \in [N]$, a confidence interval $[\texttt{LCB}_t(n), \texttt{UCB}_t(n)]$ is computed so that $r(n) \in [\texttt{LCB}_t(n), \texttt{UCB}_t(n)]$ with probability at least $1 - \delta_t$. Similarly to the proof of Theorem 2, we consider a "good event" stating that all confidence intervals where valid on a given time range before $t$,

$$\mathcal{G}_t = \left\{\forall s \in \left[\left\lfloor \frac{3t}{16}\right\rfloor, t\right], \forall n \in [N], \ r_t(n) \in [\texttt{LCB}_t(n), \texttt{UCB}_t(n)]\right\} \,.$$

It is clear that $\sum_{t=1}^{+\infty} \mathbb{P}(\bar{\mathcal{G}}_t) \leq \sum_{t=1}^{+\infty} N^2 t \delta_{\lceil \frac{3t}{16}\rceil}$, where the second union bound on $N$ comes from considering the (random) identity of the arm whose samples are used to compute the interval. The following results proves that, under $\mathcal{G}_t$, there is at least one arm in the bin $\mathcal{B}^\star$ that was played a linear number of times in $t$.

**Lemma 9** (Linear number of plays in $\mathcal{B}^\star$ under $\mathcal{G}_t$). *Under* $\mathcal{G}_t$, *there exists an arm* $n \in \mathcal{B}^\star \cup \{v_\ell^\mathcal{S}(n^\star), v_r^\mathcal{S}(n^\star)\}$ *satisfying* $m_n(3t/4) \geq \frac{t}{4K} \wedge \frac{t}{8}$, *where we call* $K = |\mathcal{S}| = \lfloor \log_{3/2}(N) \rfloor$.

*Proof.* $\mathcal{G}_t$ guarantees that any play during the interval $\left[\lfloor \frac{t}{4}\rfloor, t\right]$ (including those due to forced sampling), were decided with valid confidence intervals. Indeed, starting at $\frac{3t}{16}$ we are sure that all

forced exploration launched before that time is completed in $t/4$. Furthermore, it is also direct from the design of the algorithms that, if $r_s(n) \in [\text{LCB}_s(n), \text{UCB}_s(n)]$ and Greedy-Grid is not forced to sample the previous arm it must hold that (1) it is playing the grid, or (2) it is playing an arm in $\mathcal{B}^\star$. Indeed, no arm from a sub-optimal bin would eliminate its best neighbor.

We consider two cases. First, if an arm $n \in \mathcal{B}^\star$ was played between rounds $\lfloor \frac{t}{4} \rfloor$ and $\lfloor \frac{t}{2} \rfloor$. In that case it has collected at least $\frac{t}{8}$ samples before $t$, thanks to forced sampling. In the alternative case, the grid was played between those these two rounds, which incurs $\frac{t}{4K}$ plays of $\text{argmax}_{s \in \mathcal{S}} r(s)$ since by $\mathcal{G}$, $\text{argmax}_{s \in \mathcal{S}} r(s)$ is not eliminated when the grid is played. Then, notice that $\text{argmax}_{s \in \mathcal{S}} r(s) \subset \{n^*, v_\ell^{\mathcal{S}}(n^\star), v_r^{\mathcal{S}}(n^\star)\}$. The result follows by combining the two cases.

$\square$

Without loss of generality, we assume in the following that $K \geq 2$ (if this is not the case, just replace $K$ by $\max(K, 2)$). As a direct consequence of Lemma 9, using Theorem 1,under $\mathcal{G}_t$ there exists some constant $\beta$ and $\xi$ (coming from the bounds of the theorem multiplied by $2\sqrt{4} = 4$) such that the LCB of arm $n^\star$ satisfies

$$\forall s \in \left[\frac{3t}{4}, t\right] : \text{LCB}_s(n^\star) \geq r(n^\star) - \left\{ \beta \sqrt{K \frac{\log\left(\frac{2\lceil N\sqrt{t} \rceil}{\delta_t}\right)}{t}} + N \times \xi \left(K \frac{\log\left(\frac{2\lceil N\sqrt{t} \rceil}{\delta_t}\right)}{t}\right)^{\frac{2}{3}} \right\}. \tag{26}$$

To simplify the notation, we denote the right-hand term by $\mathcal{E}(t)$ in the rest of the proof. Using this result, we can now consider all the sub-cases presented in the theorem. We fix a sub-optimal arm $n$, and upper bound $\sum_{t=1}^{T} \mathbb{1}\{n_t = n, \mathcal{G}_t\}$ depending on the position of $n$ with respect to the grid $\mathcal{S}$. Similarly to what we did in the proof of Theorem 2, we relate the pulls due to forced sampling to actual decisions by stating that, if $n_t = n$, then there exists a round $s$ between $t - \lfloor t/4 \rfloor$ and $t$ such that GG requested a pull of arm $n$ from a "grid play" or a "greedy play".

**Case 1:** $n \in \mathcal{S}$. In that case if $n$ is pulled then, since there is no forced sampling for the arms of the grid it directly holds that

$$\text{UCB}_t(n) \geq \max_{n' \in [N]} \text{LCB}_t(n') \geq \text{LCB}_t(n^\star) \geq r(n^\star) - \mathcal{E}(t). \tag{27}$$

On the other hand, under $\mathcal{G}_t$ it holds that

$$\text{UCB}_t(n) \leq r(n) + \underbrace{\left\{ \beta \sqrt{K \frac{\log\left(\frac{2\lceil N\sqrt{t} \rceil}{\delta_t}\right)}{m_n(t)}} + N \times \xi \left(K \frac{\log\left(\frac{2\lceil N\sqrt{t} \rceil}{\delta_t}\right)}{m_n(t)}\right)^{\frac{2}{3}} \right\}}_{\mathcal{E}'(t)},$$

so pulling arm $n$ is possible only if $\mathcal{E}(t) + \mathcal{E}'(t, m_n(t)) \geq \Delta_n$, that we simplify to $2\mathcal{E}'(t, m_n(t)) \geq \Delta_n$ or $2\mathcal{E}(t) \geq \Delta_n$. Considering the second term leads to a constant problem-dependent bound, analogous to $t_{\alpha,n}$ in the proof of Theorem 2. Hence, we focus on the first term, that provide the $\log(T)$ bound.

This time, we don't know if $m_n(t)$ is large or not. This explains why we obtain a UCB-like ($\widetilde{\mathcal{O}}(\log(T))$) upper bound with this technique. We use that

$$t \leq T \Rightarrow \log\left(t \frac{2\lceil N\sqrt{t} \rceil}{\delta_t}\right) \leq \log\left(T \frac{2\lceil N\sqrt{T} \rceil}{\delta_T}\right) = \widetilde{\mathcal{O}}(\log(T)).$$

Then, similarly to proof of Theorem 2, we mitigate the asymptotic scaling in $N$ by noticing that if $m_n(t) = \Omega(N^6)$ then $\frac{N}{m_n(t)^{\frac{1}{6}}}$ simplifies. In that case, we obtain a sub-Gaussian confidence interval, and similarly to the analysis of UCB [3] we obtain that arm $n$ is only pulled at most $\widetilde{\mathcal{O}}\left(\frac{\log(T)}{\Delta_n^2} \vee N^6\right)$ times with high probability. This is the first part of the result for $n \in \mathcal{S}$.

We then use another analysis to derive the constant problem-dependent bound. We remark that, when the confidence intervals are valid, the best arm between $v_\ell^{\mathcal{S}}(n^\star)$ and $v_r^{\mathcal{S}}(n^\star)$ can only be eliminated by an arm $i_t^\star \in \mathcal{B}^\star$. By design of the algorithm (exploiting the unimodality assumption), it furthermore holds that if $i_t^\star \in \mathcal{B}^\star$ and those two arms are eliminated then GG does not play on the grid. Hence, the constant bound in this case comes from upper bounding the time required for this event to happen under $\mathcal{G}_t$. If $v_\ell^{\mathcal{S}}(n^\star)$ is not eliminated it must hold that $\texttt{UCB}_t(v_l^{\mathcal{S}}(n^\star)) \geq \texttt{LCB}_t(n^\star)$ (if $n \leq n^\star$). Furthermore, Lemma 9 also guarantees that

$$\texttt{UCB}_t(v_\ell(n^\star)) \leq r(v_\ell(n^\star)) + \mathcal{E}(t) \ ,$$

therefore the event that $v_\ell(n^\star)$ is not eliminated is only possible if $2\mathcal{E}(t) \geq \Delta_{v_\ell(n^\star)}^2$. We can then use Lemma 7, following the same as in the proof of Theorem 2 right after (23) and (24). When $T$ is large enough, the derivation provides the scaling $\widetilde{\mathcal{O}}\left(\frac{1}{\Delta_{v_\ell^{\mathcal{S}}(n^\star)}^2}\right)$. We can then can follow the same steps for $v_r^{\mathcal{S}}(n^\star)$, and obtain $\widetilde{\mathcal{O}}\left(\frac{1}{\Delta_{v_r^{\mathcal{S}}(n^\star)}^2}\right)$. Furthermore, it is clear by analogy with the proof of Theorem 2 that for small values of $T$ the upper bounds have to be multiplied by a $N^2$ factor. Finally, we can remark that if $n < n^\star$ the first bound is used, while the second is used for $n > n^\star$. This concludes the derivation of the upper bound for $n \in \mathcal{S}$.

**Case 2:** $n \notin \mathcal{S}$, $\mathcal{B}(n) \neq \mathcal{B}^\star$. We prove that this case is actually impossible under the good event, which explains the surprising constant upper bound independent of any gap. Indeed, if $n \notin \mathcal{S}$ is played, then it must hold that its right and left neighbors in the grid are eliminated. Since $\mathcal{B}(n) \neq \mathcal{B}^\star$ then at least one of them has a reward at least as good as any arm in $\mathcal{B}(n)$. However, if playing arm $n$ was possible under $\mathcal{G}_t$ it would hold that

$$\begin{aligned}
\exists \ell \in \mathcal{B}(n) : \quad r(\ell) &\geq \texttt{LCB}_t(\ell) \\
&> \max\{\texttt{UCB}_t(v_\ell^{\mathcal{S}}(n)), \texttt{UCB}_t(v_r^{\mathcal{S}}(n))\} \\
&\geq \max\{r(v_\ell^{\mathcal{S}}(n)), r(v_r^{\mathcal{S}}(n))\} \geq r(\ell) \ ,
\end{aligned}$$

which is a contradiction due to the strict inequality in the second line. Hence, the number of time such arm $n$ is played is simply upper bounded by $\sum_{t=1}^{+\infty} \mathbb{P}(\bar{\mathcal{G}}_t)$, which is (by design) bounded by a universal constant.

**Case 3:** $n \notin \mathcal{S}$, $\mathcal{B}(n) = \mathcal{B}^\star$ We use that $n_t = n$ implies that $n_s = n$ *due to a greedy play* at some round $s \in [3t/4, t]$. Under $\mathcal{G}_t$, we can thus directly use (26), and obtain that if $2\mathcal{E}(t) \leq \Delta_n^2$ this event is not possible anymore. Using the same derivation as in the other cases (involving Lemma 7), we obtain the upper bound scaling in $\mathcal{O}\left(\frac{1}{\Delta_n^2}\right)$ for $T$ large enough, and by $\mathcal{O}\left(\frac{N^2}{\Delta_n^2}\right)$ in general.

$\square$

## C.5 Regret of Greedy-Grid adapted for non-unimodal rewards

In this section we develop the result presented in Section 3, regarding the adaptation of Greedy-Grid in the case when the reward function is no longer assumed to be unimodal. We recall that the adaptation consists in simplifying the definition of the set $\mathcal{C}_t$ in Algorithm 2 by

$$\mathcal{C}_t = \{s \in \mathcal{S}, U_s \geq L_{i_t^*}\} \ .$$

We call the resulting algorithm GG-NU, for *Greedy-Grid Non Unimodal*, to differentiate it from the original version of GG introduced in the paper. In the following, we formalize the upper bound of the regret of GG-NU, and discuss how this result is obtained by adapting the proof of Theorem 3 from the previous section.

**Theorem 4** (Regret of GG-NU). *Suppose that* GG *is tuned with confidence level* $\delta_t = \frac{1}{N^2 t^3}$, *and* $\alpha = 1/4$. *Then, for any* $T \in \mathbb{N}$ *it holds that*

$$\mathcal{R}_T = \widetilde{\mathcal{O}}_N\left(\sum_{n \in \mathcal{B}^\star} \frac{1}{\Delta_n} + \sum_{n \in \mathcal{S}} \frac{\log(T)}{\Delta_n}\right) \ .$$

*Additionally, it holds that* $\mathcal{R}_T = \widetilde{\mathcal{O}}\left(\sqrt{(K + |\mathcal{B}^\star|)T}\right)$, *for* $K = \lfloor \log_{3/2}(N) \rfloor$.

*Proof.* The proof follows the exact same steps as the proof of Theorem 3 presented in Appendix C.4. To adapt the arguments to GG-NU, we first remark that it suffices to identify which part of the proof uses the definition of the set $\mathcal{C}_t$. We then find that this is the case when analyzing the *Case 1* in the proof, namely the regret caused by sub-optimal arms in the grid $|\mathcal{S}|$. More precisely, to obtain the bound of Theorem 3 for GG we provide two simultaneously valid upper bounds: a logarithmic $(\log(T))$ upper bound with a reasoning akin to the standard UCB analysis, and then a constant upper bound that carefully leverages the definition of $\mathcal{C}_t$. We easily verify that the steps for the logarithmic bound remain valid with the new definition of $\mathcal{C}_t$, while the second bound clearly does not hold. This completes the adaptation of Theorem 3 (for GG) into Theorem 4 (for GG-NU). □

# D  Experiments

All the code for the experiments is written in Python. We use Matplotlib for plotting [18], Numpy [15] for numerical computing and Scipy [31] for scientific and technical computing. Scipy and Numpy are distributed under the BSD 3-Clause, and Matplotlib is distribution under BSD-style license. Theses licenses allow free use, modification, and distribution of the library. All the experiments were conducted on a single standard laptop, with an execution time shorter than 24 hours.

In a first simulation, we consider a coalition of size $N = 100$ and a competition of size $p = 4$. At each timestep $t$, the algorithm decide a number of bidders $n_t$ to send to the auction and the values of all bidders (coalition and competition) $\mathbf{v} \in \{0, 1\}^{n_t + p}$ are sampled according to $\mathcal{B}(0.05)$. With probability $\frac{n_t}{n_t + p}$, the reward $\mathbf{v}_{(1)} - \mathbf{v}_{(2)}$ is received and $\mathbf{v}_{(1)}$ is observed. The (pseudo) regret at time $t$ is then computed as the sum of reward obtained up to time $t$. The simulation above is repeated 20 times with random seeds and the mean value across seeds is reported as the expected regret $\mathcal{R}(t)$ in Figure 3. Error bars represent the first and the last decile.

In this simulation, the parameters are chosen to allow for having a significant number of players while keeping a gap $\Delta$ large enough (about $2 \times 10^{-4}$) to be able to observe logarithmic regrets for the baselines. LG practically outperforms other approaches by a large gap. The two algorithms that ignore the structure (UCB and EXP3) end up exhibiting a worse regret than LG, OSUB and GG, which is expected. However GG has a much higher regret than LG and only outperform UCB and EXP3 for horizons greater than $10^5$, when it starts to eliminate points from the logarithmic grid $\mathcal{S}$. Indeed, due to the explicit use of concentration bounds in the algorithm, which multiplicative constants are not optimized for practical implementations, GG does not practically reach the constant regret regime in the horizon of these simulations.

To illustrate the practical performance of LG, we perform additional simulations which are identical to the first one except for the parameters $N$, $p$, and the distribution of value that are set according to Table 2. The results are plot in Figure 4 where it is shown that LG reaches the constant regret regime after only a couple hundreds or thousands time steps while the other algorithms are still in the transient linear regime.

Table 2: Configuration of additional experiments presented in Figure 4.

| Position | $N$ | $p$ | Value distribution | $n^*$ | $\Delta$ |
|---|---|---|---|---|---|
| Top | 5 | 2 | $\mathcal{B}eta(a = 0.35, b = 0.63)$ | 3 | $5 \times 10^{-5}$ |
| Middle | 5 | 2 | $Trunc.\ Exp(0, 1)$ | 3 | $4 \times 10^{-4}$ |
| Bottom | 20 | 4 | $Trunc.\ Exp(0, 1)$ | 5 | $3 \times 10^{-5}$ |

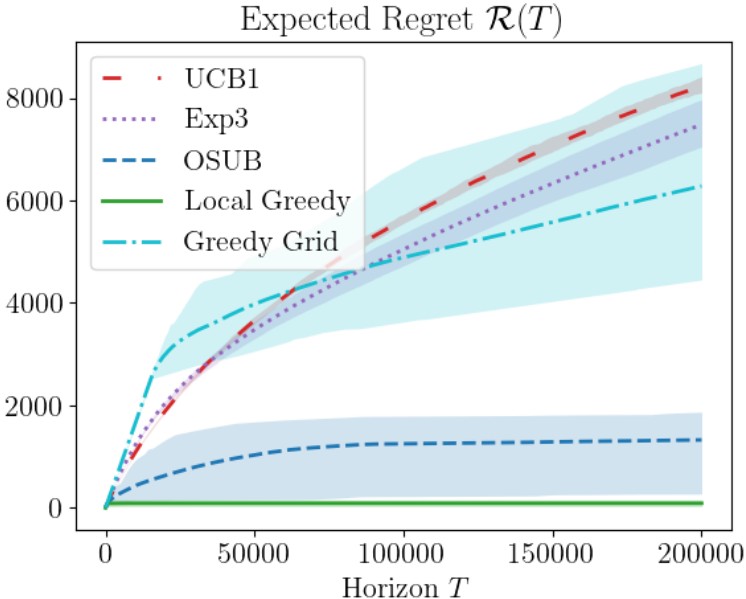

Figure 3: An empirical illustration of Table 1 with simulations in the following setting: values are distributed according to $\mathcal{B}(0.05)$, $N = 100$ and $p = 4$. We benchmark LG and GG (this paper), OSUB [9], UCB [3] and EXP3 [4] in terms of $\mathcal{R}(T)$ computed over 20 trajectories. Error bars represent the first and last decile.

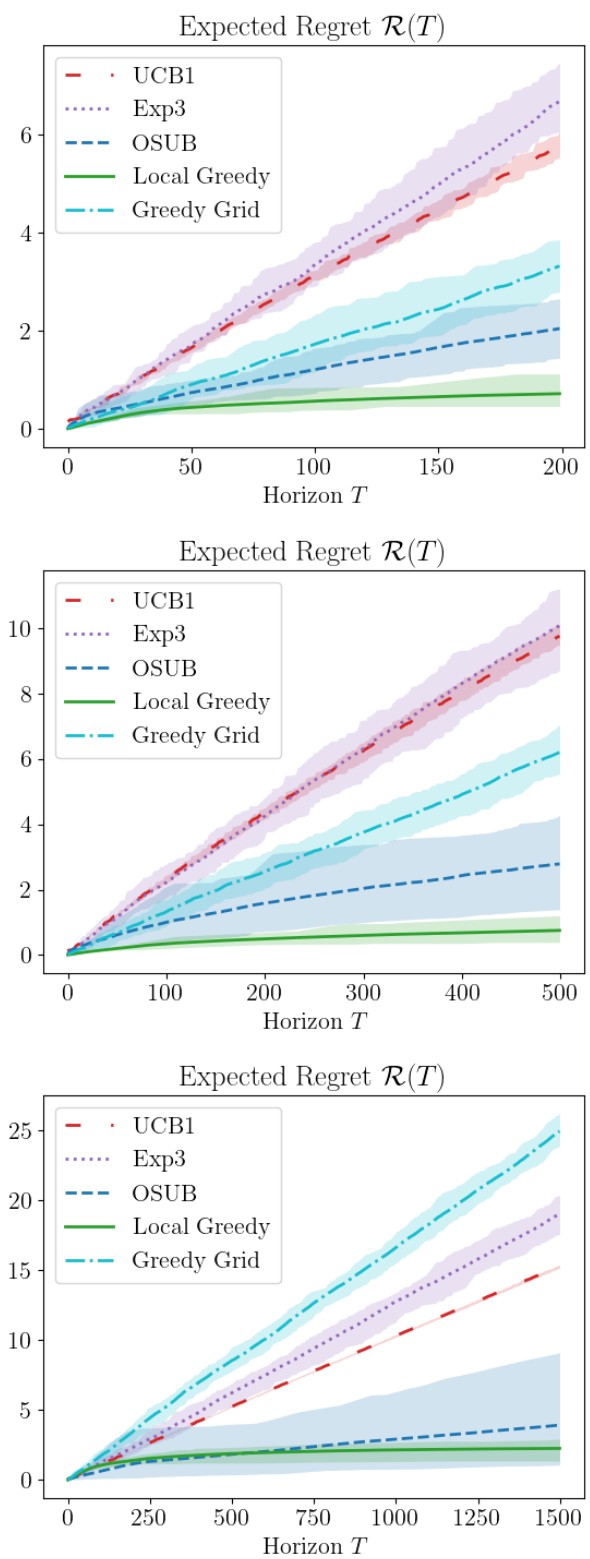

Figure 4: Illustration of additional experiments. Details of parameters are provided in Table 2.

**Complexity analysis** The time-complexity of both `GG` and `LG` mainly comes from the computation of reward estimates. They are computed by replacing the integral in Equation (5) by a Riemann sum

with $\lfloor N\sqrt{T} \rfloor$ terms (the reasoning behind the number of terms needed is the same as in the proof of Theorem 1. Therefore, whenever reward estimates of a neighborhood of size $\mathcal{O}(N)$ is needed, it costs $\mathcal{O}(N^2\sqrt{T})$ operations. Note that during forced exploration steps, reward estimates are not needed and therefore the associated cost is not paid. The total time complexity therefore depends on the number of times reward estimates are needed which itself depends on the trajectory. However, our algorithms could be modified to guarantee that reward estimates are needed at most $\mathcal{O}(\log(T))$, times for instance by only performing updates at the end of phases of exponentially increasing length. This would lead to a mean complexity per iteration of $\mathcal{O}(N\log(T))$, and similar theoretical guarantees by a slight adaptation of the analysis. This would reduce the burden of using of incorporating the structure in the algorithm, compared to the $\mathcal{O}(N)$ cost of the baselines (which is even $\mathcal{O}(1)$ for OSUB).

# E   Broader Impact

The collection of user data should be carried out with the preservation of user privacy in mind. This issue is at the forefront of recent, ongoing developments, such as the European Union's General Data Protection Regulation (GDPR) and the California Consumer Privacy Act (CCPA).

In online advertising, maintaining privacy presents new challenges, as decisions must be made without complete access to user data.

This paper tackles one such challenge by detailing Multi-Armed Bandit (MAB) algorithms that Demand Side Platforms (DSPs) can use to determine the number of ad campaigns that should partake in the repeated auction for ad placements, without the need for prior knowledge of each campaign's value. This study is therefore a step towards the realization of practical user privacy. It is important to note that the assumptions we make require that the actions of the DSP have a limited impact on the market, which should be carefully verified in practical applications.

