# OpenReview forum: "Optimizing the coalition gain in Online Auctions with Greedy Structured Bandits"
_NeurIPS.cc/2024/Conference — NeurIPS 2024 poster_

### Official Review · Reviewer_RP8o · 2024-07-04

**Soundness:** 3
**Presentation:** 3
**Contribution:** 3
**Rating:** 6
**Confidence:** 4

**Summary:**

The paper explores strategies for maximizing the cumulative reward in repeated second-price auctions, where agents draw their values from an unknown distribution. The problem is modeled as a multi-armed bandit (MAB) scenario with structured arms, where the decision-maker selects a subset of agents to compete in each auction to optimize the coalition's total gain. Two novel algorithms, Local-Greedy (LG) and Greedy-Grid (GG), are introduced, both achieving constant problem-dependent regret. LG, while practical and outperforming GG in experiments, avoids reliance on confidence intervals. GG, on the other hand, offers problem-independent guarantees and better theoretical performance. The paper also presents new concentration bounds and theoretical guarantees for these algorithms, positioning them as significant advancements in the field of online auction optimization using structured bandit approaches.

**Strengths:**

The paper introduces a highly original approach to optimizing online auctions using structured bandits, specifically through the Local-Greedy (LG) and Greedy-Grid (GG) algorithms. This novel problem formulation merges auction theory with multi-armed bandit (MAB) frameworks, particularly for online display advertising. By focusing on bidder coalitions and the inherent structure of the reward function, the paper overcomes some limitations of previous work and offers a fresh perspective on auction optimization.

The theoretical contributions are substantial and well-supported. The authors rigorously derive regret bounds and introduce new concentration bounds crucial for performance guarantees. The detailed proofs and theoretical analysis in the appendices demonstrate the robustness of the methods, providing a solid foundation for understanding the algorithms' behavior under various conditions. This rigor ensures the results are both novel and reliable.

The paper's clarity is another strength. The authors explain complex concepts clearly and logically, progressing from problem formulation to algorithm development, theoretical analysis, and experimental validation. Key ideas and assumptions are stated clearly, and the algorithms are described in detail, making the methodology easy to follow. Illustrative examples and tables, like the comparison of regret guarantees, further enhance readability and understanding.

The work's significance is evident in its application to online auction optimization in online display advertising, a multi-billion-dollar industry. Improving auction algorithms can lead to significant economic benefits. By addressing privacy constraints and the need for efficient ad placement strategies, the proposed methods can greatly enhance the performance of demand-side platforms (DSPs) in real-world settings. The theoretical advancements also contribute to the broader field of MAB research, especially in structured and unimodal bandits.

**Weaknesses:**

One significant weakness of the paper is the limited experimental validation. The authors primarily use synthetic data, which does not fully demonstrate the practical applicability of the algorithms in real-world online auctions. To strengthen the practical relevance, the methods should be validated using actual auction data. Collaborating with industry partners or using publicly available datasets would improve the experimental section.

Another concern is the narrow comparison with existing methods. The paper compares LG and GG algorithms with a few standard multi-armed bandit algorithms like UCB, EXP3, and OSUB. However, there are many relevant algorithms in the literature, such as privacy-preserving bandits, that could provide a more comprehensive benchmarking. Including more sophisticated models that consider bidder heterogeneity and dynamic bidding strategies would offer a more rigorous evaluation.

The theoretical assumptions, particularly the unimodality of the reward function and the i.i.d. nature of bidder values, may be overly restrictive. These assumptions simplify the analysis but may not hold in real-world settings where bidder values can be correlated and vary over time. Relaxing these assumptions and exploring their impact would provide a more realistic evaluation. This could involve theoretical extensions or empirical investigations to understand the sensitivity of the algorithms.

The discussion on practical implementation lacks depth. While the paper provides theoretical guarantees and some empirical results, it does not sufficiently address the computational complexity and scalability of the algorithms. In real-world applications, especially with thousands of bidders, efficiency is crucial. A detailed analysis of computational costs and potential optimizations or approximations would benefit practitioners implementing these algorithms in large-scale systems.

Lastly, the paper could benefit from a detailed exploration of parameter selection and sensitivity analysis. The performance of both LG and GG algorithms depends on several parameters, such as the exploration parameter $\alpha$ and the confidence levels $\delta$. However, there is no thorough analysis of how to choose these parameters or their sensitivity to different settings. A comprehensive parameter study, including guidelines for selecting appropriate values, would enhance the usability of the proposed methods. This could involve both theoretical insights and empirical evaluations to provide a clear understanding of the parameter space.

**Questions:**

1. Can you provide more details on why you chose to use synthetic data for your experiments? If you have plans to include real-world data in future work, could you outline how this might impact your findings?
2. How sensitive are your algorithms (LG and GG) to the choice of parameters like the exploration parameter $\alpha$ and the confidence levels $\delta$? It would be helpful to understand the range of values for these parameters that ensure robust performance across different scenarios.
3. The paper assumes unimodality of the reward function and i.i.d. bidder values. Can you provide more empirical or theoretical justification for these assumptions? Are there any real-world scenarios where these assumptions might not hold, and how would that impact your algorithms' performance?
4. Could you provide a detailed analysis of the computational complexity of LG and GG? Specifically, how do these algorithms scale with an increasing number of bidders and auctions? Are there practical optimizations or approximations that could reduce the computational burden without significantly impacting performance?
5. The paper focuses on second-price auctions. How would your algorithms need to be adapted for other types of auctions, such as first-price or generalized second-price auctions? Are there specific challenges or advantages in these other auction settings?
6. The paper introduces new concentration bounds. Could you elaborate on the derivation of these bounds and how they compare to traditional Hoeffding bounds in practice? Are there specific scenarios where your bounds provide significant advantages?

**Limitations:**

The authors have provided a thorough theoretical foundation and practical implications for their algorithms, but there are areas where addressing limitations and potential societal impacts could be improved. They mention some limitations in their theoretical assumptions, such as the unimodality of the reward function and the i.i.d. nature of bidder values, but these could be more explicitly discussed in terms of their practical implications. For instance, real-world scenarios where these assumptions might not hold should be identified, and potential mitigation strategies should be outlined. Additionally, while the paper focuses on improving algorithmic performance, the broader societal impacts, such as how these algorithms might affect market fairness or competition in online advertising, are not discussed.

---

> ### Author Rebuttal · Authors · 2024-08-05
>
> Thank you for your detailed review and questions. As you noticed, our main contributions consist in providing theoretically sound algorithms for the problem that we introduce, under some assumptions that we motivate. Our simulations allow to check
> that our theoretical insights are valid when these assumptions are satisfied.
>
> We believe that an extensive
> empirical evaluation would definitely be a nice addition but prefer to carry it
> in future work. Your questions about the use of synthetic data, the comparison with
> existing methods, the unimodality and iid assumptions, the computational
> complexity and hyper-parameter selection are answered in the next paragraphs.
>
> (W1/Q1) As we are considering a bandit setting, we cannot use a fixed dataset
> for experiments as the experiment needs to give the reward *depending on*
> the action played by the algorithm. This can only be done by interacting with a
> real-world live system or with a simulator. Even if we had the possibility to
> run our algorithms on a live real-world system, it would make the experiment not
> reproducible. In the end, only the choice of simulator makes the experiments
> reproducible. We could have adapted to our problem the simulator available at
> https://github.com/amzn/auction-gym. Note however that this simulator also relies
> on simple distributions for the values (ex: lognormal for the one mentioned
> above), which is similar to what we do.
>
> (W2) We believe that our comparison with existing
> methods is fair. It is not clear to us that the algorithms mentioned by the reviewer apply to our setting. First, while it emerges from new privacy-preserving regulations in online advertisements, our problem is different from settings where the bandit algorithm itself has to enforce some notion of privacy. Secondly, we emphasize that in our setting the individual bidders bid their values and thus there is no need to use an algorithm to specifically learn how to bid.
>
> (W3/Q3) We answer about unimodality in the general comment above. The
> case where bidders value evolve over time would be an
> interesting extension, that is discussed in the answer of Reviewer TZVE (Q2).
>
> (W4/Q4) In the revision, we will provide a complexity analysis in Appendix D. In both GG and LG, the most costly part in term of time-complexity is the computation of reward estimates. They are computed by replacing the integral in Equation (5) by a Riemann sum with $\lceil{N \sqrt{T}\rceil - 1}$ terms (the reasoning behind the number of terms needed is the same as in the proof of Theorem 1, precisely line L664). Therefore, whenever reward estimates of a neighborhood of size $O(N)$ is needed, it cost $O(N^2 \sqrt{T})$ operations. Note that during forced exploration steps, reward estimates are not needed and therefore the associated cost is not paid. The total time complexity therefore depends on the number of times reward estimates are needed which itself depends on the trajectory.
> However, our algorithms could be modified to guarantee that reward estimates are needed at most $\log(T)$ times for instance by only performing updates at the end of a phase of exponentially increasing length. This would lead to a mean complexity per iteration of $O(N \log(T))$.
>
> (W5/Q2) Regarding the $\delta_t$, as standard with confidence-based algorithms we want to set it as small as possible so that the theoretical guarantees hold. This is the case if the sum in l.890 converges, which is guaranteed by the tuning that we propose in Theorem 3 (more precisely, this term becomes $\sum_{t\geq 1} t^{-2}$). Regarding $\alpha$: (1) for LG we use $\alpha=1/(\log_{3/2}(N)+1)$, which is strongly supported by the analysis l.860-862. We will add this tuning in the statement of Theorem 2 in the revision. (2) for GG our analysis holds if $\alpha$ is smaller than $1/2$, Theorem 3 suggests $\alpha=\frac1{4}$ . The tuning of $\alpha$ does not seem crucial for its performance because it is only used after the best neighborhood is identified with large probability.
>
> (Q5) In any online setting, first-price auctions are inherently much harder to study due to the game-theoretical aspects coming from the fact that the different bidders can adapt their bidding strategy at every time step. With the symmetry assumption, a Nash equilibrium exists and is known if all bidders bid separately, but it requires the *knowledge* of the value distribution, which is exactly what the decision maker is trying to learn. Due to that, even if a stationary regime exists asymptotically, it is very hard to provide any guarantee on finite-time behavior. Generalized second-price auctions are also non-truthful and would present similar challenges, on top of requiring to extend the setting to selling multiple items at each time.
>
> (Q6) The novel concentration bounds presented in Theorem 1 are one of the major technical contributions of the paper. We provide a sketch of the proof l.161-174 of the paper and a detailed proof in Appendix B, where we first present auxiliary results before providing the complete proof. In particular, in Appendix B.1.2 we detail the concentration inequalities that we use to obtain tailored concentration bounds for each part of the integral defining the expected reward (Eq. 5). First, Theorem 1 is essential to analyze the concentration of simple estimates of $r(l)$ based on samples from an arm $n \neq l$. Indeed, a standard Hoeffding bound can only be used to estimate $r(n)$ with rewards from arm $n$ only. Hence, none of our theoretical guarantees can be obtained without Theorem 1. Furthermore, in the proof of Theorem 1 significant effort was required to remove the $n$ factor coming from the definition of the reward (Eq. 5). As discussed in l 183-185, this result cannot be obtained with a standard Hoeffding bound.
>
> (Limitations paragraph) We will include some of the reviewers' suggestions in Appendix E.

---

> > ### Comment · Reviewer_RP8o · 2024-08-13
> >
> > Thank you to the authors for their detailed rebuttal, which has addressed most of my concerns. I will maintain my positive rating.

---

### Official Review · Reviewer_Ja3N · 2024-07-09

**Soundness:** 3
**Presentation:** 3
**Contribution:** 3
**Rating:** 5
**Confidence:** 3

**Summary:**

This paper studies the problem of repeatedly selecting the number of agents to form a coalition against the environment to maximize the cumulative reward in second-price auctions. Specifically, the paper supposes that all bidders are identical with unknown value distribution $F$, and in each round $t$, $n_t$ bidders out of $N$ bidders would be chosen, and truthfully bid against $p$ bidders. The goal is to maximize the expected reward function $r(n_t)$. Under the assumption that $r(n)$ is an unimodal function of $n$, the paper first presents an estimation of $r$ via powers. Under this component, the paper presents two algorithms: Local-Greedy ($\mathtt{LG}$) and Greedy-Grid ($\mathtt{GG}$), which both utilize the segmentation structure of the estimation result given above. The difference is that Local-Greedy searches locally, while Greedy-Grid does an iterative elimination on segments. The regret of Local-Greedy is bounded by a problem-dependent constant, and the regret of greedy-grid is $\tilde{O}(\sqrt{T})$ (I am not sure if I am correct here).

**Strengths:**

The problem studied in this work is quite interesting and has some practical implications. I very much like the estimation result as given by the paper, which utilizes the structure of the problem well and provides strong insights for designing searching algorithms with low regret. I believe this is the major contribution of the work and is solid. In fact, with the sectioned structure of the estimation, the two proposed algorithms are natural (but not naive). Overall, the paper is also well written.

**Weaknesses:**

In my opinion, the major weakness of the work is that all the results are supported by the assumption that $r(n)$ is an unimodal function of $n$. In my intuition, when $r(n)$ is multi-modal, both proposed algorithms may stop searching in a sub-optimal peak. Please correct me if I misinterpret. I wonder if such a mis-stopping can be prevented by doing the search more "patiently".

Also, in the experiment part, it seems that the Local-Greedy algorithm always enjoys a better performance than Greedy-Grid, and Greedy-Grid seems to perform worse than OSUB. This makes the theoretical regret results of Greedy-Grid weaker.

**Questions:**

1. Can the authors provide intuitions on what will happen to the proposed algorithms when the function $r(n)$ is not unimodal?
2. Is it prevalent that Greedy-Grid performs worse than OSUB in practice? Could the authors explain intuitively/theoretically why Local-Greedy works better than Greedy-Grid?

**Limitations:**

The authors have addressed the limitations of their work.

---

> ### Author Rebuttal · Authors · 2024-08-05
>
> Thank you for your careful review. First, regarding the last
> sentence of your summary, we want to precise that for Greedy-Grid (GG) the regret bound we
> obtain is **the minimum** between
> $\sqrt{(\log_2(N)+|\mathcal{B^\star}|)T}$ and a problem-dependent constant
> (independent of $T$). We answer your question about unimodality above, in a
> general comment for all reviewers. Below, we answer your second question about
> the comparison between the performance of Local-Greedy (LG)/OSUB and Greedy-Grid
> (GG).
>
> First, we refer to the discussion at the end of the paper (l.306-337) for a detailed comparison of the theoretical results obtained with our two algorithms. We recall that both algorithms admit constant problem-dependent regret (see Theorem 2 and 3), but the scaling of the constant is better for GG, and furthermore permits to derive $O(\sqrt{T})$ problem-independent guarantees (which we cannot obtain for LG due to the *local* gap in the bound). Hence, from a theoretical perspective Greedy-Grid enjoys better guarantees than Local-Greedy. However, our experiments indeed show that Local-Greedy performs better in practice. In our opinion, there are several reasons that might explain this gap between theory and practice.
>
> A first reason is that the scaling in the worst local gap in the analysis of Local Greedy (Theorem 2) might be conservative: this gap can be paid in a scenario combining bad initialization (very far from the optimal arm and with flattest part of the reward function on the path towards the optimal neighborhood) and bad luck (a ''plausibly maximal'' number of steps is taken to move in the good direction). It is likely that in practice this scenario is quite rare, and thus not seen in our experiments. Although the analysis of LG might be made slightly less conservative (see l.322-327 for a discussion), it seems impossible to completely remove some local gaps from the analysis of Local-Greedy in the worst case.
>
> Furthermore, the analysis do not capture the fact that in some cases
> Local-Greedy might also be lucky and start playing in the optimal neighborhood
> very fast. On the other hand, this situation cannot happen for GG, which
> requires enough statistical evidence to eliminate all sub-optimal neighborhoods.
> While this step in GG is the reason for its improved theoretical guarantees, it might have an empirical cost because GG is ``always cautious'' compared to LG. Lastly, we discuss l.334-337 that another reason for that practical gap might also be that implementing LG does not require to compute confidence intervals, contrarily to GG. Hence, it is possible that the confidence intervals of Theorem 1 can be tightened, which would directly benefit to the performance of Greedy-Grid by accelerating the search for the optimal neighborhood. We leave potential refinements of the bound presented in Theorem 1 for future work. Additionally, we believe that these intuitions also hold when comparing GG and OSUB, from which LG is strongly inspired. We believe that the comparison between LG and OSUB justifies the interest of our approach from a practical perspective.
>
> We will complete the "Experimental results" paragraph l.338 with the parts of this discussion that are not already presented in the previous paragraph l.306-337, in order
> to provide more intuitions about why LG works better than GG in our
> experiments.
> We believe that analyzing those two algorithms in our paper provides a good
> overview of what is possible to achieve: GG has the most appealing theoretical
> guarantees but is outperformed by LG in our experiments, suggesting LG
> is a better choice in practice.
> Whether LG can be modified to achieve the theoretical guarantees we obtain
> for GG is an interesting but
> challenging open problem.

---

> > ### Comment · Reviewer_Ja3N · 2024-08-12
> >
> > Thanks for your kind response! I'm keeping my score positive.

---

### Official Review · Reviewer_TZVE · 2024-07-14

**Soundness:** 3
**Presentation:** 3
**Contribution:** 3
**Rating:** 6
**Confidence:** 4

**Summary:**

This paper studies repeated second-price auctions with ex-ante bidder coalition. There are two groups of bidders, one of size N and one of size p. At each period $t$, a decision maker can choose $n_t$ out of the N bidders to compete with the other p bidders in an auction. Crucially, the decision maker chooses the bidders without knowing their values (due to privacy concern). The decision maker aims to maximize the total expected reward (computed according to the second-price auction rule). This problem reduces to a multi-arm bandit problem with a structured reward function, where each possible number $1\le n \le N$ is an arm. The authors design two algorithms, Local-Greedy and Greedy-Grid, to solve this problem with constant-in-T problem-dependent regret and $\sqrt T$ problem-independent regret. Techniques include a non-trivial concentration bound for the reward functions and adaptations of the unimodal bandit algorithm OSUB.

**Strengths:**

(S1) [Significance] This work is well motivated. It is motivated by: (1) The interesting observation that the demand-side-platforms (DSP) on online advertising auctions can coordinate the bidders to maximize the total gain; (2) Due to privacy concern, DSP can only choose coalition without seeing the bidders' values. In particular, (2) naturally turns the problem into a multi-armed-bandit problem with small arm space and structured reward function, which is an interesting connection.

(S2) [Significance & Quality] The authors find algorithms that achieve problem-dependent regret bounds that are *independent of T*, which is a surprising result, especially when compared to the UCB1 algorithm with problem-dependent log(T) regret bounds. A problem-independent regret bound in the order of $O(\sqrt{\log N + |\mathcal B^*|T})$ that is better than UCB1 and EXP3 is also given. These results are impressive and the underlying analysis are non-trivial.

(S3) [Quality] Experiments are provided to validate the advantage of the proposed algorithms, which is good.

(S4) [Clarify] The writing is very clear overall. While the full proofs of the results are involved, the authors nicely summarize the main ideas and discuss interesting aspects in the proofs, including potential improvements in the analysis. This is very good.

(S5) [Significance] The proposed future directions are interesting, and can potentially lead to some follow-up works.

**Weaknesses:**

(W1) The assumption that the reward function $r$ is unimodal (Assumption 2) feels restricted. Although the authors briefly discuss how to handle non-unimodal functions in Line 311 - 313, no formal result is given.

(W2) The assumption of symmetric bidders also feels restricted.  When bidders are asymmetric, the decision maker can choose either (1) a subset of bidders or (2) a uniformly random subset of n bidders from the set of N bidders to compete with the other p bidders in an auction.  Can the authors provide some discussions about asymmetric bidders?

**Questions:**

(Q1) In addition to the maximum realized value, can the decision maker also observe the reward (maximum value - payment) when the auction is won? Line 44 says yes, but Line 62 says no. If the reward (equivalently, the second highest value) is also observed when the auction is won, can the results be improved?

(Q2) Can the results be generalized to the case where the bidders' value distributions change over time (but the bidders are still symmetric within each period)?

(Q3) What if bidders are asymmetric?  See (W2).

**Limitations:**

**Suggestions:**

(1) Capitalize "Coalition Gain" in title.

(2) Line 2: "their values"

---

> ### Author Rebuttal · Authors · 2024-08-05
>
> Thank you for your detailed review and suggestions, as well as for appreciating the key contributions of our work. Regarding the unimodality assumption (W1), we answer in a general comment for all reviewers. Regarding the symmetry of bidders, see the paragraph (Q3) below.
>
> (Q1) We apologize for the inconsistency, we will correct line 44. Indeed, our analysis is conducted by assuming that only the maximum reward is observed. However, we could use the same procedure presented in Section 2.2 if the sequence of second prices (price paid by the winner) $\\overline{S}\_{k} = (s\_{k,1}, \\dots, s\_{k, m_k})$ was observed instead of $\\overline{W}_k$. The cumulative distribution function of observations would become $G_k: x \\in [0,1] \\mapsto (k+p) F(x)^{k+p-1} - (k+p-1)F(x)^{k+p}$, and thus it is also possible to estimate any power $F^\\ell$ from $\\overline{S}_k$ via a suitable inversion formula. Furthermore, the same could be done if both first and second prices were observed, but the inversion would be more intricate because the joint distribution of $\\overline{W}_k$ and $\\overline{S}_k$ should be considered. This might allow a slight improvement of the multiplicative constants of the result, but probably at the cost of more intricate computation. We will add this discussion at the end of Section 2.2.
>
>
> (Q2) This question is indeed particularly relevant in real-world applications, thank you. First, the progressive change of distribution can imply a change in optimal arm, but also that previous observations progressively induce biased estimators. Hence, we believe that our algorithms might require some adaptation to tackle a non-stationary value distribution. Since our problem is framed as a structured MAB, it might be natural to look for inspiration in the rich literature on non-stationary MAB problems. In this literature, the general idea is to mitigate this bias by forgetting some of the past observations, either passively (with a sliding window or discount factor, see e.g. [1]) or actively (with change-point detectors, see e.g. [2]).
> Then, the analysis of the *dynamic regret* can be conducted by assuming a property of the non-stationarity, such as an upper bound on a number of changes or on the total amplitude of variations (we could imagine a budget on $\sum_{t=1}^T ||F_t-F_{t+1}||_\infty$ for instance). We believe that our algorithms can be adapted to include these ideas, and the practitioners' choice would certainly be to opt for a simple sliding window mechanism. We leave the formalization of this adaptation and its analysis for future work. We suspect that the structure of our problem might imply interesting non-trivial properties, for instance by making it easier to detect changes in the reward function because changes in $F$ can be detected when sampling any arm.
>
>
> (Q3) The assumption of symmetric bidders is commonly used in the auction
> literature (see e.g chapter 2 of [3]) to build understanding of problems that couldn't be solved in the
> full generality of arbitrary bidders.
>
> Following your comment in (W2), if we considered asymmetric bidder the action
> space would include all possible combinations of players from the coalition,
> making the problem combinatorial. In addition, the estimation of the reward
> function with different value distributions becomes very intricate, so overall
> this setting might be too difficult to actually exploit the structure of the
> problem.
>
> We believe that studying the symmetric case is a necessary step to unlock addressing more complex settings in future work, such as coalitions built out of several groups of similar bidders.
>
>
> [1] Garivier, Aurélien, and Eric Moulines. "On upper-confidence bound policies for non-stationary bandit problems." arXiv preprint arXiv:0805.3415 (2008).
>
> [2] Besson, Lilian, et al. "Efficient change-point detection for tackling piecewise-stationary bandits." Journal of Machine Learning Research 23.77 (2022): 1-40.
>
> [3] Krishna, Vijay. Auction theory. Academic press, 2009.

---

> ### Comment · Reviewer_TZVE · 2024-08-11
> **Keep rating 6**
>
> I read the authors' response.  All of my concerns except for W1 (unimodal $r$) are resolved.  For W1, the authors' response on how to relax the unimodal assumption looks promising.  Nevertheless, I understand that the authors cannot provide the full formal argument during rebuttal, which makes it difficult to verify the correctness of this argument -- the devil is always in the details.  With that said, I am still positive with this paper because I think other strengths outweigh this minor weakness, so I keep recommending weak accept.

---

### Author Rebuttal · Authors · 2024-08-05

**General comment**  We want to thank all the reviewers for their careful examination of our paper. We appreciate that all reviewers seem enthusiastic about the problem that we introduce, the algorithms that we propose and the theoretical contributions presented in our work.


**Unimodality of the reward function**  Following their questions, we now propose a common response for all reviewers about the unimodality assumption presented in the paper (Assumption 2).

First, we recall (Lemma 2 in the paper) that unimodality holds for several usual families of distributions.

Secondly, by carefully following the proofs of Theorem 2 and 3 we can remark that our theoretical results actually hold under a slightly milder assumption: it suffices that the estimation neighborhood of every sub-optimal arm contains a better rewarding arm, that is
$n\\neq n^\\star \\Rightarrow \\exists n' \\in \\mathcal{V}(n):\\; r(n')>r(n)$   with the notation from the paper. This assumption seems easier to satisfy, especially if $p$ is large, by construction of the neighborhoods. For instance, if all arms form a single estimation neighborhood then unimodality is not needed at all.

Then, we can also consider the case where even this assumption does not hold. As pointed out by Reviewer TZVE, we provide l.311-313 of the paper a simple and sound way to adapt Greedy-Grid in that case: in Algorithm 2, (line 4 in the "for" loop) the set $\mathcal{C}_t$
can be redefined as follows:  $\\mathcal{C}\_t= \\{s \\in \\mathcal{S}:\\; U_s \\geq L\_\{i_t^*\} \\} $  .

Indeed, the formulation of $\\mathcal{C}\_t$ in Greedy-Grid (Algorithm 2) exploits unimodality by allowing some arms with large confidence intervals to remain eliminated if any arm on their path towards the empirical best LCB is eliminated. The consequence of that change for the theoretical guarantees is straightforward: in Theorem 3 the term $\\sum_{n \\in \\mathcal{S}} \\frac{\\log(T)}{\\Delta_n} \\wedge C_n$ (where here we use $C_n$ to denote the constant term in the theorem) becomes $\\sum_{n \\in \\mathcal{S}} \\frac{\\log(T)}{\\Delta_n}$. Furthermore, the problem-independent guarantees remain identical. Hence, **without unimodality a slight adaptation of GG achieves logaritmic regret on the subset of arms in the grid $\\mathcal{S}$, and still obtain $\\sqrt{(\\log_2(N)+|\\mathcal{B^\\star}|)T}$ problem-independent regret**. This is better than what standard UCB would obtain: logarithmic regret for **all $N$ arms**, and $\\sqrt{NT}$ for problem-independent bounds. Following the remark of Reviewer TZVE, we will formalize this discussion and the adaptation of Greedy-Grid in a dedicated appendix, that we will reference at l.313 of the paper.

---

### Decision · Program_Chairs · 2024-09-25

**Decision:**

Accept (poster)

**Comment:**

The paper reports a study on repeated second-price auctions formulated as a structured bandit problem. The reviewers are in general positive towards the paper, acknowledging that the problem formulation, the technical results and analyses have novelties. The reviewers also pointed out a number of issues for the authors to address. I recommend acceptance to the paper, and encourage the authors to make a thorough revision of the paper addressing all the issues raised by the reviewers.